# Larval precompetency and settlement behaviour in 25 Indo-Pacific coral species

Carly J. Randall [1,2 ✉], Christine Giuliano [1], Briony Stephenson[1], Taylor N. Whitman[1,2], Cathie A. Page[1], Eric A. Treml[3], Murray Logan[1] & Andrew P. Negri[1]

Knowledge of coral larval precompetency periods and maximum competency windows is fundamental to understanding coral population dynamics, informing biogeography and connectivity patterns, and predicting reef recovery following disturbances. Yet for many species, estimates of these early-life history metrics are scarce and vary widely. Furthermore, settlement cues for many taxa are not known despite consequences to habitat selection. Here we performed a comprehensive experimental time-series investigation of larval settlement behaviour, for 25 Indo-Pacific broadcast-spawning species. To investigate the duration of precompetency, improve predictions of the competency windows, and compare settlement responses within and amongst species, we completed replicated and repeated 24-hour assays that exposed larvae to five common settlement cues. Our study revealed that larval competency in some broadcast-spawning species begins as early as two days post fertilization, but that the precompetency period varies within and between species from about two to six days, with consequences for local retention and population connectivity. We also found that larvae of some species are competent to settle beyond 70 days old and display complex temporal settlement behaviour, challenging the assumption that competency gradually wanes over time and adding to the evidence that larval longevity can support genetic connectivity and long-distance dispersal. Using these data, we grouped coral taxa by short, mid and long precompetency periods, and identified their preferred settlement cues. Taken together, these results inform our understanding of larval dynamics across a broad range of coral species and can be applied to investigations of population dynamics, connectivity, and reef recovery.

[1] Australian Institute of Marine Science, Townsville, QLD, Australia. [2] AIMS@JCU, Townsville, QLD, Australia. [3] Australian Institute of Marine Science, Perth, WA, Australia. ✉email: c.randall@aims.gov.au

Many coral ecological processes, including recovery, metapopulation persistence, adaptation, and range expansion, are dependent on population connectivity through dispersal by pelagic, free-swimming larvae[1–3]. The degree to which local populations are interconnected is influenced by ocean currents and larval biology, including settlement competency characteristics, pelagic larval durations (PLDs), and larval settlement behaviour[4–6]. Yet despite the fundamentally important role that larval biology plays in driving metapopulation dynamics at ecological time scales, biological data are lacking for key coral life-history parameters that are known to shape connectivity, for the vast majority of species. For species with well-described life-history parameters, there is significant variability within and between cohorts and amongst species, driven by diversity in reproductive mode, method of symbiont acquisition, and other larval characteristics[7,8]. In addition, species-specific plasticity in precompetency has been documented in response to changing environmental conditions[9–11], adding to the known variance in these parameters. Understanding this variability, and the degree of plasticity in these early life-history metrics, is critical to predicting coral persistence and adaptation under climate change, and to implementing appropriate ecosystem management and conservation actions.

Both the timing of the onset of settlement competency (i.e. the precompetency period)[8,9,12] and the competency duration[2,13,14] have consequences for local larval retention, dispersal (population connectivity)[2,5,6,11,15,16], and gene flow (genetic connectivity)[3,17–19], and are central to understanding biogeography and evolution[1]. The precompetency period influences local retention and demographics, particularly over short (annual to decadal) time-scales[5,8]. For example, Figueiredo et al. 2013 integrated coral larval competency models with estimates of particle retention around reefs and found that species with shorter precompetency periods were more likely to recruit locally. Indeed, rates of larval retention (as modelled as particles) around reefs are often a similar order to the rates at which competency is acquired[8,20]. Cetina-Heredia and Connolly[20] modelled larval retention times for a variety of reef types using a hydrodynamic model and found that mean water residence times were 0.5–5.6 days. Small differences in the precompetency period, therefore, can have significant consequences for local retention and dispersal. Similarly, in a theoretical coral population connectivity model, Treml et al.[5] found that local retention was strongly related to the length of the precompetency period. Unfortunately, the duration of precompetency has only been estimated for around a dozen broadcast-spawning coral species and ranges 6-fold, from <2 days in *Goniastrea retiformis*[8] to 12 days in *Acropora lordhowensis*[13] (Table S1). Previous evidence also suggests that, for marine invertebrate larvae, propagule size may be an important predictor of precompetency duration[8,21,22]. Yet whether this relationship is consistent across taxa and can be used to estimate such biological parameters requires further exploration. This variability undoubtedly has consequences for local retention, dispersal dynamics and metapopulation connectivity. Therefore, empirically quantifying precompetency across a broad range of coral taxa is necessary to predict how ecological processes on reefs will respond to shifting disturbance regimes and accelerating climate change[10,23,24].

Studies of larval longevity (i.e., maximum pelagic larval duration (PLD)) and dispersal are similarly rare due to the inherent difficulties in following and sampling sub-mm-sized larvae in the ocean over vast distances, and the presence of confounding factors such as environmental variability and habitat heterogeneity. Consequently, estimated PLDs vary widely, ranging 10-fold, from 20 to 200+ days (Table S1). Of those studies that have experimentally measured PLD, many assume that settlement competency is maintained as long as larvae are alive without confirming

successful larval settlement at the end of the PLD[3] but see refs. [2,11,15]. Yet, Connolly and Baird[2] reported that after reaching peak competency, there was an exponential decline in competency in five coral species (4–13 days), reaching near 0 by 60–120 days. However, this loss of competency varied five-fold amongst species, again illustrating high inherent variability in larval biology across Scleractinia. Actively swimming larvae require energy to function (including for locomotion and settlement); however, endogenous lipid energy reserves can decline quickly during the PLD[3,14,25]. For example, larval dry weight decreased by 50% within 30 days of spawning in *Acropora tenuis*[26], and total lipids declined by 64% within 30 days in *Goniastrea retiformis* larvae[25]. Graham et al.[3] found that larvae ceased swimming after the first month, with energy reserves reaching critically low levels 100 days after spawning in five broadcasting species. Thus, while the survival of lecithotrophic larvae may be supported for months, the energetic costs of settlement (i.e. attachment and metamorphosis)[27] may not be possible for the entire PLD. Therefore, it is necessary to understand the capacity of larvae to settle following these extended PLDs and describe the competency window within the PLD[2].

While the cues that initiate larval attachment and metamorphosis of Acroporid corals have been well studied (e.g. refs. [28–32]), much less is known about the specificity of settlement cues for the other ~20 genera that collectively dominate coral cover on the Great Barrier Reef (GBR), such as *Porites* and *Goniastrea* (reviewed in ref. [32]). Evidence to date suggests that preferred settlement cues vary considerably amongst species[32–34], and contribute to variation in coral community structure and zonation[28,35]. Furthermore, whether a species is considered a 'generalist settler' (i.e. one that responds to many cues), or a 'specialist settler' (i.e. a species that requires a specific inductive cue) and whether sensitivity to settlement cues changes throughout the competency window (i.e. the 'desperate larval hypothesis' *sensu*[36]) remain largely unexplored in corals. Indeed, most studies that investigate settlement cue preferences do so during peak competency and don't assess potential temporal variability. Investigating larval settlement behaviour through time can dramatically improve our understanding of these dynamics.

Population connectivity and the retention of larvae drive local-scale demography and system-level metapopulation dynamics[37–39] but are also essential for natural recovery following disturbances[40,41]. A population that is connected to several upstream larval sources will have a greater capacity to recover quickly following a devastating event (e.g., ship grounding, cyclone or mass-bleaching event). Similarly, local populations that sustain patchy or modest mortality following disturbances have a greater recovery potential when locally sourced larvae are retained[38]. In addition, it has been recently highlighted that variability per se in larval connectivity at a metapopulation level may also bolster persistence[42]. A greater understanding of early life-history characteristics, and their variability, is essential in determining natural reef recovery rates.

In response to recent declines in coral cover[43,44] and recruitment on the GBR[45], efforts to develop restoration interventions to aid recovery and enhance heat tolerance are also underway[46–49]. Many of these interventions rely on the sexual propagation of corals in aquaculture systems and depend on knowledge of larval biology[50,51]. One such method is coral seeding, which aims to deliver newly settled coral polyps (i.e. 'spat') to the reef via the deployment of seeding devices[46,52–54]. To incorporate meaningful biodiversity in reef restoration efforts, and to restore the appropriate coral species for a given reef site, knowledge to reliably and predictably spawn, rear, and settle a diversity of species is required. Therefore, identifying when settlement can commence,

for how long larvae are competent to settle, and what cues to use to induce settlement across diverse taxa are research priorities.

Here we investigated the in vitro larval settlement behaviour of 25 GBR coral species across three years of mass-spawning (Table S2). We performed replicated 24-h settlement assays starting during precompetency and tracked larval settlement up to 77 days post-spawning, to (i) define the precompetency period, (ii) test settlement responses to a variety of common cues, and (iii) improve our predictions of settlement competency windows. We also explored whether egg diameter, as a proxy for larval nutrition, correlated with time to competency.

## Results

**Precompetency**. The average estimated time to settlement competency (TC50) across species was 4.0 days after fertilization (DAF), while the shortest TC50 was found for *Goniastrea*

*retiformis* at 2.1 DAF and the longest precompetency was estimated for *Acropora austera* at 6.2 DAF (Fig. 1; Fig. 2b; Table 1). 14% of species were classified as having 'short' precompetency periods (i.e. <3 days) and most were Merulinids. 'Mid' precompetency duration (3–5 days) was most common (71% of species) and included taxa from five families (Diploastraeidae, Lobophyllidae, Poritidae, Merulinidae and Acroporidae). Only three species, all of the genus *Acropora*, were classified as having 'long' precompetency periods (i.e. >5 days) (Table 1). For some species, the TC50 estimate differed significantly amongst cues (Fig. 1; Supplementary Code).

A significant positive relationship was detected between oocyte diameter and TC50 ($p = 0.011$, adj $R^2 = 0.26$; Fig. 2a). All species with short precompetency periods had oocytes less than ~450 μm average diameter, while oocytes of all species that had long precompetency periods were ~500 μm or larger. However, the

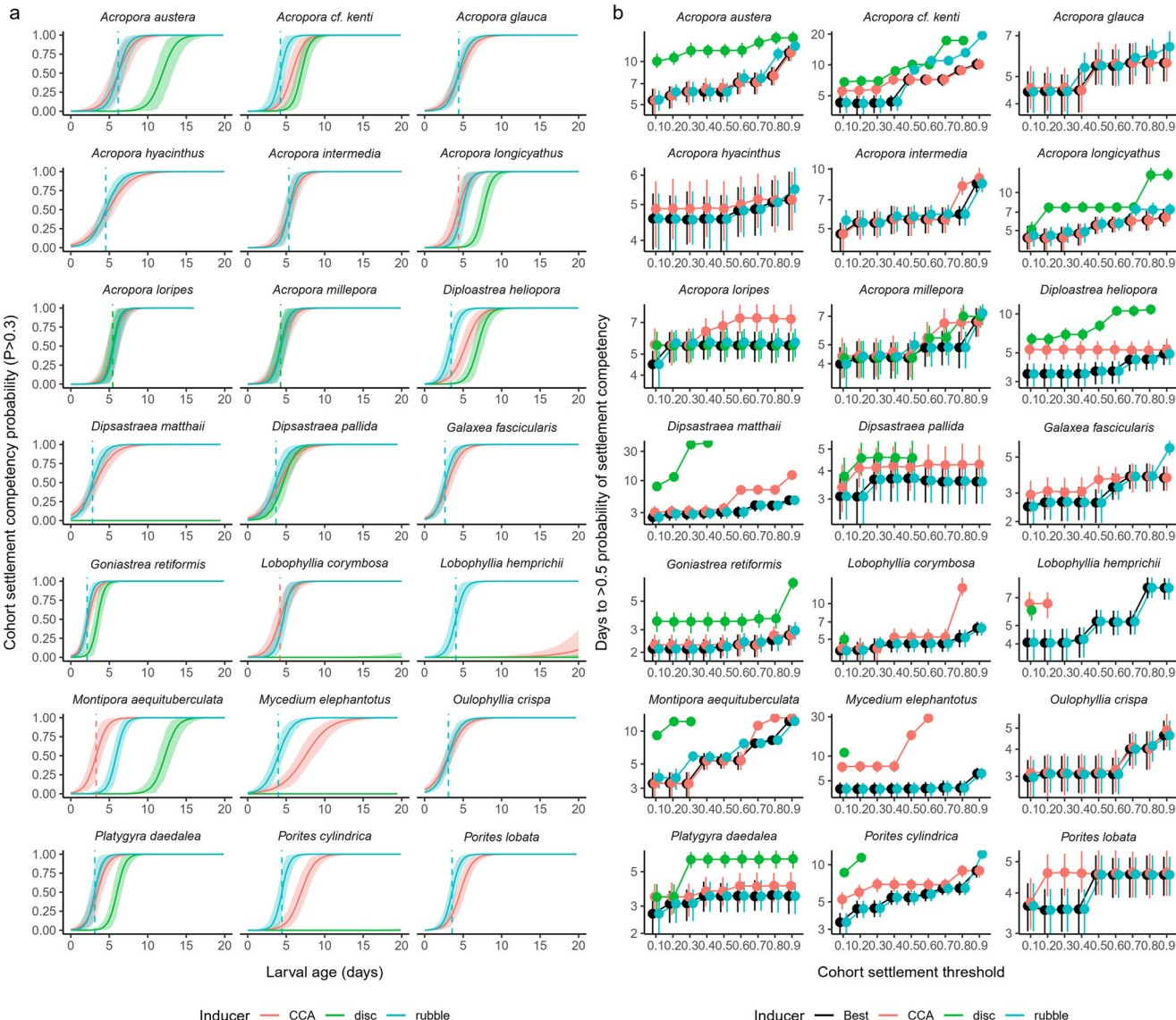

**Fig. 1 Modelled time to settlement competency. a** Partial plots of the modelled cohort settlement probabilities (of exceeding a 0.3 threshold) against larval age for each cue and species. Solid curves represent the modelled posterior medians while the ribbons represent the associated 95% credible intervals around the estimates. Dashed lines correspond with the earliest TC50 estimate for that species and are coloured based on the cue. **b** Modelled days to >0.5 settlement probability against cohort settlement threshold (0.1–0.9), conditional on cue treatment for each species. 'Best' shows the data for whichever cue predicted the earliest time to settlement for a given cohort settlement threshold. Points represent posterior medians and error bars represent 95% credible intervals. Estimated medians are truncated to exclude durations that are greater than the extent of the experiments. Note the variable *y*-axes.

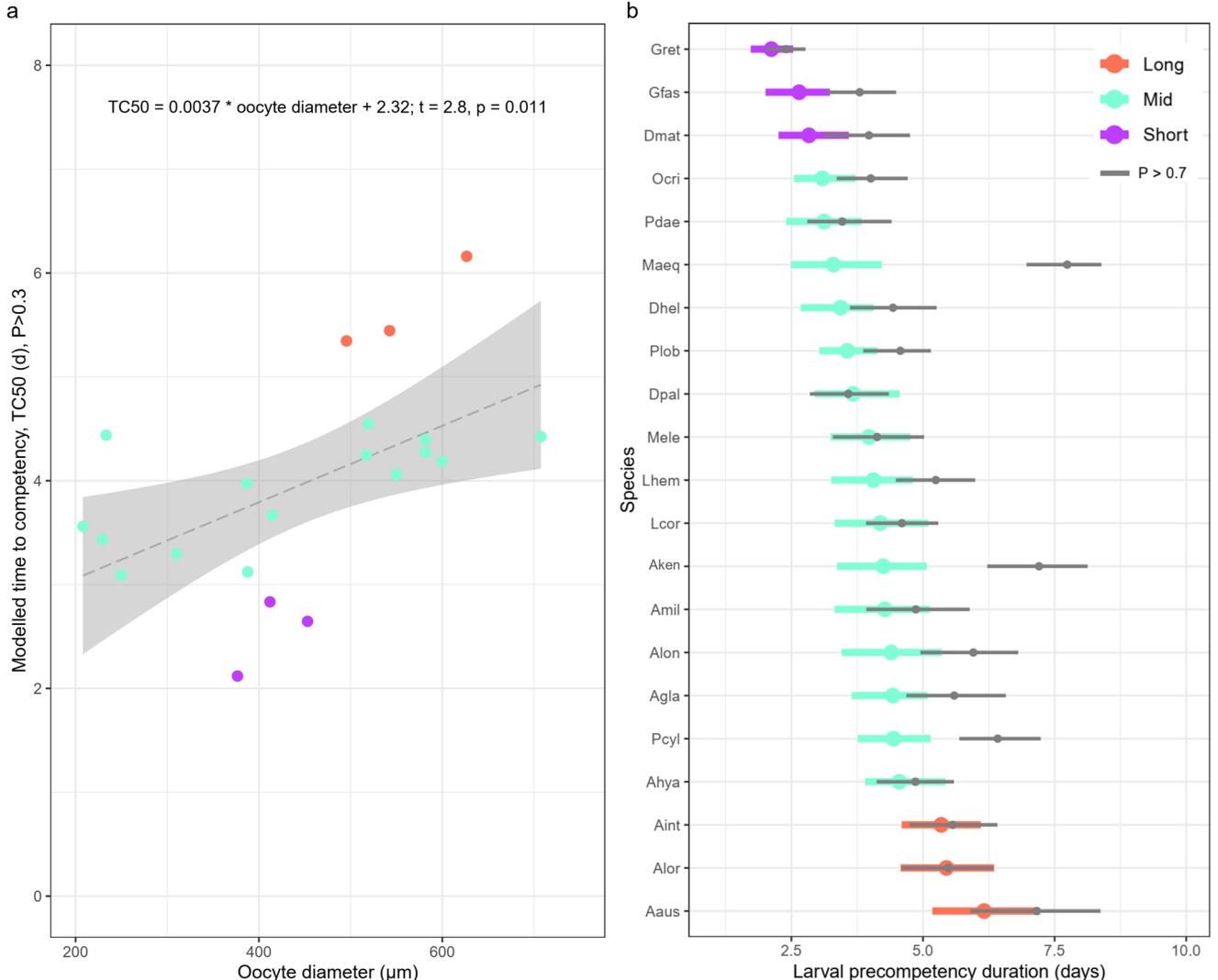

**Fig. 2 Time to settlement competency by oocyte diameter and species. a** Relationship between oocyte diameter (μm) and the modelled time to competency (TC50) for 21 coral species (Table 1). Colours represent precompetency duration where short <3 days, mid-3–5 days, long >5 days. Grey dashed line represents the linear regression with confidence intervals. **b** Modelled TC50 at P > 0.3 for each species, ordered and coloured by precompetency classification of short, mid, or long, with the TC50 estimate at P > 0.7 overlaid in grey. Estimates for the cue that provided the earliest TC50 estimate are P > 0.3 and are shown for both thresholds. Points indicate TC50 estimates and bars represent the range of the credible interval around each estimate. Species abbreviations as per.

greatest range in oocyte diameter was detected in mid-precompetency taxa and included those with the smallest (*Porites lobata*) and largest (*Acropora glauca*) oocytes.

TC50 estimates were also calculated using threshold settlement values from 0.1 to 0.9 (10–90%) in increments of 0.1. For many species, TC50 estimates were consistent across much of the range in thresholds, particularly for the most effective cue (see 'best' treatment in Figs. 1b and 2b). However, species such as *M. aequituberculata* and *P. cylindrica* had gradually increasing TC50 estimates as the threshold increased.

**Competency duration and competency patterns through time.** Most species remained competent to settle until the final time-point of testing, often beyond 30 days (Fig. 3). Three of four species tested past 70 days remained capable of settlement above 30% (*A. hyacinthus*, *D. matthaii*, and *D. pallida*), and *Oulophyllia crispa* settled at its final timepoint of 59 days.

Larval settlement also fluctuated through time (Fig. 3) with peaks and troughs that were often consistent amongst cues within

and between species. A complex, multimodal settlement pattern was also observed for all five species tested beyond 40 days (i.e. *A. hyacinthus*, *D. matthaii*, *D. pallida*, *O. crispa* and *M. elephanto-tus*). In some cases (i.e. *D. matthaii*, *O. crispa*) the number of inductive cues increased during the second peak (Fig. 3). Some indiscriminate (i.e. 'spontaneous') settlement in the negative control was also observed throughout the study, although it was rare (Fig. 3).

**Inter- and intra-specific patterns in response to settlement cues.** The most effective cues overall were reef rubble and CCA, both of which cued every species to settle significantly better than the control (Table 2). Biofilm discs elicited settlement responses in ~80% of species, while the CCA extract and peptide worked for 74% and 65% of species, respectively.

Conditioned rubble was the most effective cue for most species and typically elicited over 80% settlement during peak competency (Fig. 3; Table 2; Supplementary Data 1). The most effective settlement cue within the Acroporidae varied by species. The

**Table 1 Age to settlement competency.**

| Family | Species (abbreviation) | Modelled (TC50) (days) | Lower CI | Upper CI | Cue inducing earliest TC50 | Pre-competency period ranking | Average oocyte diameter (μm) ref |
|---|---|---|---|---|---|---|---|
| Acroporidae | Acropora austera (Aaus) | 6.16 | 5.18 | 7.22 | Rubble | Long | 627 *[120,] |
| | Acropora glauca (Agla) | 4.42 | 3.64 | 5.09 | Rubble | Mid | 708[120] |
| | Acropora hyacinthus (Ahya) | 4.55 | 3.90 | 5.43 | Rubble | Mid | 520[120–122] |
| | Acropora intermedia (Aint) | 5.35 | 4.59 | 6.10 | Rubble | Long | 496[120,121] |
| | Acropora longicyathus (Alon) | 4.40 | 3.45 | 5.35 | CCA | Mid | 582[120,*] |
| | Acropora loripes (Alor) | 5.44 | 4.58 | 6.35 | Disc | Long | 543[120,*] |
| | Acropora millepora (Amil) | 4.27 | 3.32 | 5.13 | Disc | Mid | 582[120,123,*] |
| | Acropora cf. kenti [tenuis] (Aken) | 4.24 | 3.36 | 5.07 | Rubble | Mid | 517[120] |
| | Montipora aequituberculata (Maeq) | 3.30 | 2.49 | 4.21 | CCA | Mid | 310[121,*] |
| Diploastraeidae | Diploastrea heliopora (Dhel) | 3.43 | 2.68 | 4.06 | Rubble | Mid | 230[121,124] |
| Euphylliidae | Galaxea fascicularis (Gfas) | 2.65 | 2.01 | 3.23 | Rubble | Short | 453[121] |
| Lobophylliidae | Lobophyllia corymbosa (Lcor) | 4.19 | 3.32 | 5.10 | CCA | Mid | 600* |
| | Lobophyllia hemprichii (Lhem) | 4.06 | 3.26 | 4.81 | Rubble | Mid | 550[122,123] |
| Merulinidae | Dipsastraea matthaii (Dmat) | 2.83 | 2.26 | 3.59 | Rubble | Short | 412* |
| | Dipsastraea pallida (Dpal) | 3.67 | 2.94 | 4.56 | Rubble | Mid | 414[121,123] |
| | Goniastrea retiformis (Gret) | 2.12 | 1.73 | 2.53 | Rubble | Short | 377[87,121–123] |
| | Mycedium elephantotus (Mele) | 3.97 | 3.24 | 4.76 | Rubble | Mid | 387[121,*,] |
| | Oulophyllia crispa (Ocri) | 3.09 | 2.55 | 3.72 | Rubble | Mid | 250[124] |
| | Platygyra daedalea (Pdae) | 3.12 | 2.40 | 3.83 | Rubble | Mid | 388[121] |
| Poritidae | Porites cylindrica (Pcyl) | 4.44 | 3.76 | 5.15 | Rubble | Mid | 233* |
| | Porites lobata (Plob) | 3.56 | 3.03 | 4.13 | Rubble | Mid | 208[121] |

Modelled age (days) to settlement competency, where the cohort is more likely than not (TC50) to have exceeded a 30% settlement threshold, for 21 coral species. The cue (i.e. CCA, rubble or disc) that provided the earliest TC50 estimate at a 30% threshold is identified. Upper and lower credible intervals (CI) around the TC50 estimates are presented. Precompetency periods are ranked by short (<3 days), mid (3–5 days), and long (>5 days) duration. CCA = Crustose coralline algae from the *Porolithon* spp. complex. *reference for oocyte diameter is C.J. Randall, personal observation.

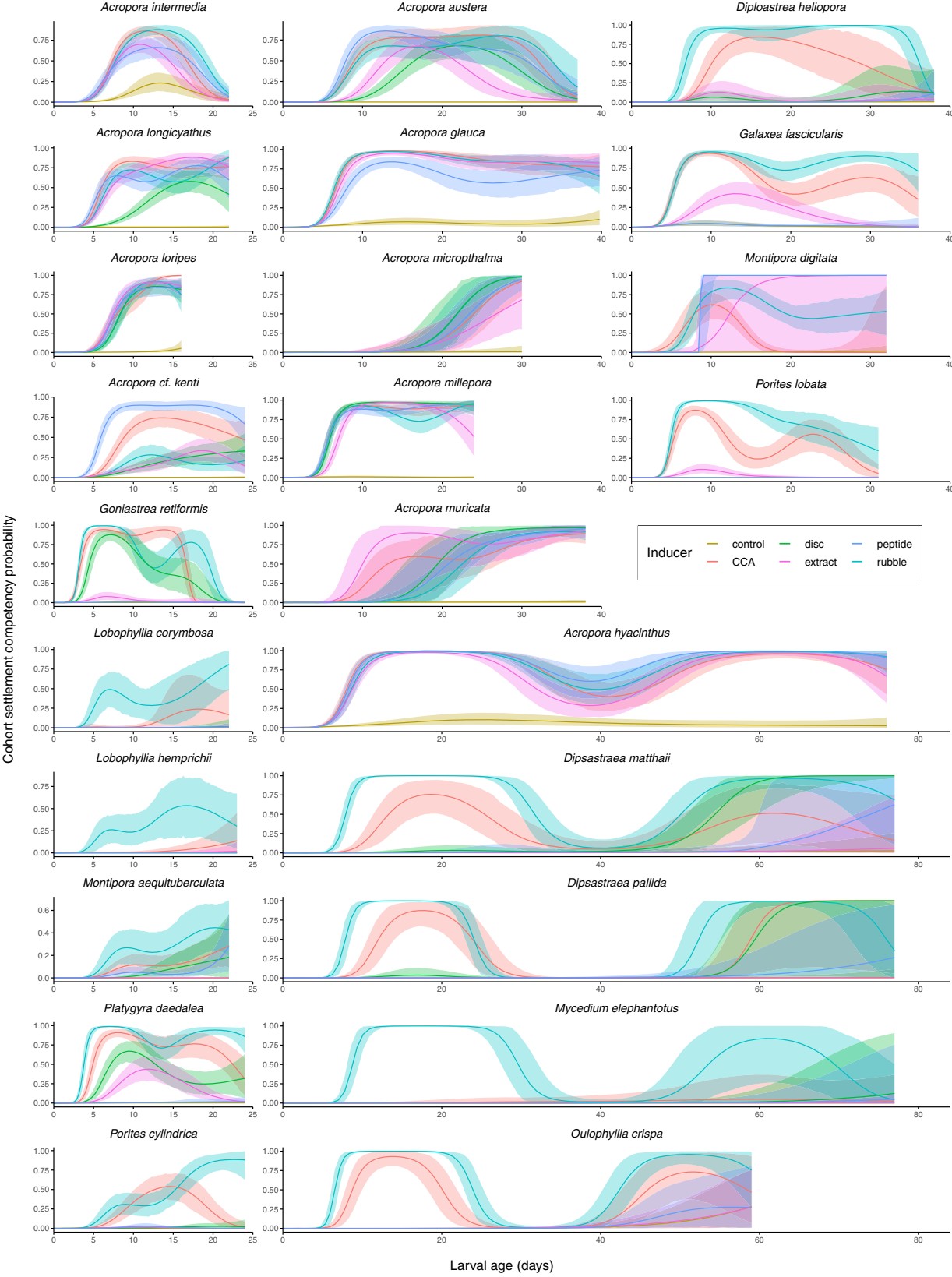

**Fig. 3 Settlement through time by species.** Generalized additive hierarchical models of the probability of settlement through time for each cue tested, for each species. Colours represent settlement inducer treatments. Solid lines represent posterior medians and ribbons represent associated 95% credible intervals. Note that *x*-axes (larval age in days) vary amongst species but are scaled equivalently.

**Table 2 Settlement cue effectiveness.**

| Family | Species | Cue ranking |
|---|---|---|
| Acroporidae | *Acropora austera* | CCA = Rubble = Peptide > Disc = Extract > Control |
| | *Acropora glauca* | CCA = Rubble = Extract > Peptide > Control |
| | *Acropora hyacinthus* | Peptide > Rubble = CCA > Extract > Control |
| | *Acropora intermedia* | Rubble > CCA = Peptide > Extract > Control |
| | *Acropora longicyathus* | CCA = Extract > Peptide = Rubble > Disc > Control |
| | *Acropora loripes* | Extract = CCA = Peptide = Rubble = Disc > Control |
| | *Acropora micropthalma* | Disc = Rubble = Peptide = CCA = Extract > Control |
| | *Acropora millepora* | Disc = CCA = Peptide > Rubble = Extract > Control |
| | *Acropora muricata* | Extract = CCA = Disc = Peptide = Rubble > Control |
| | *Acropora cf. kenti (tenuis)* | Peptide > CCA > Disc = Extract = Rubble > Control |
| | *Montipora aequituberculata* | Rubble > CCA = Disc = Peptide > Control = Extract |
| | *Montipora digitata* | Peptide = Extract = Rubble > CCA > Control |
| Diploastraeidae | *Diploastrea heliopora* | Rubble > CCA > Disc = Extract > Peptide > Control |
| Euphylliidae | *Galaxea fascicularis* | Rubble > CCA > Extract > Peptide = Control |
| Lobophylliidae | *Lobophyllia corymbosa* | Rubble > CCA > Disc = Peptide = Extract = Control |
| | *Lobophyllia hemprichii* | Rubble > CCA > Extract = Disc = Control = Peptide |
| Merulinidae | *Dipsastraea matthaii* | Rubble > Disc = CCA > Peptide = Control = Extract |
| | *Dipsastraea pallida* | Rubble > CCA = Disc = Peptide = Control > Extract |
| | *Goniastrea retiformis* | Rubble = CCA > Disc = Extract > Peptide > Control |
| | *Mycedium elephantotus* | Rubble > CCA = Disc = Peptide = Extract = Control |
| | *Oulophyllia crispa* | Rubble > CCA > Peptide = Extract = Control |
| | *Platygyra daedalea* | Rubble > CCA > Disc > Extract = Peptide > Control |
| Pachyseridae | *Pachyseris speciosa* | Rubble = CCA > Control = Peptide = Disc = Extract |
| Poritidae | *Porites cylindrica* | Rubble > CCA > Disc = Peptide = Extract > Control |
| | *Porites lobata* | Rubble > CCA > Extract > Peptide = Control |

Overall ranking of settlement cue effectiveness for each species, from strongest to weakest, based on estimated pairwise differences in the area under the curve between inducer treatments, from a Bayesian generalized additive hierarchical model (Fig. 3). CCA = crustose coralline algae from the *Porolithon* spp. complex. Note that "disc" was not tested for all species.

GLW-amide peptide cued all Acroporidae spp. to settle significantly better than the control but was ineffective in all other families (Table 2).

## Discussion

We performed an experimental investigation of larval settlement behaviour for a broad taxonomic cross-section of Indo-Pacific coral species and found that precompetency periods ranged from about 2 to 6 days and duration increased with egg size. We also confirmed that extended competency windows (+70 days) are possible for at least some species, and identified novel and complex temporal dynamics in settlement behaviour during the competency window that may facilitate long-distance dispersal success. These patterns challenge the long-held assumption that settlement competency gradually wanes over time, with likely significant implications for population connectivity and meta-population dynamics. Finally, we demonstrated that reef rubble was a broadly effective settlement cue, providing a starting point for further investigations into cultivating potential universal inducers.

Precompetency durations ranged 3-fold across taxa, with the Merulinid *Goniastrea retiformis* on one end of the spectrum at 2.1 DAF, and the staghorn *Acropora austera* on the other end of the spectrum at 6.1 DAF. Figueiredo et al.[8] modelled time to competency for six broadcasting coral species and found that estimates ranged from 1.4 to 3.8 days. Of the six species Figueiredo et al.[8] tested, three were again tested in this study and our estimates were remarkably similar: we estimated precompetency periods of 2.1, 4.3 and 3.1 days, compared to 1.4, 3.5 and 2.9 days for *G. retiformis*, *A. millepora* and *P. daedalea*, respectively. Using these precompetency periods, Figueiredo et al.[8] modelled the proportion of larvae retained, based on water residence times on the GBR, and found that an increase in precompetency of as little as 1 day led to significantly reduced local larval retention. Therefore, our results, which quantified precompetencies ranging

from 2.1 to 6.1 days, suggest that larval retention should vary widely amongst coral taxa. For example, if local-scale currents are moving at 10 cm s$^{-1}$, a 4-day increase in precompetency would result in an additional 35 km of transport.

The rate of development through ontogeny is often size-dependent, influenced by the average amount of energy needed to create cells (i.e. more rapid development for cells of less mass)[55]. Egg size governs larval size, with smaller conspecific marine invertebrate larvae reaching competency earlier[56]. In corals, egg size has been positively correlated with time to motility[8] and propagule size has been positively correlated with precompetency duration in brooding corals[57]. Our results support this relationship and expand this to more taxa and reproductive strategies, indicating that larger embryos take longer to reach settlement competency. The significant relationship between oocyte diameter and TC50, therefore, may be used to estimate TC50 for other taxa for which competency data are unknown (Fig. 2a).

High variation in precompetency allowed us to classify species as having either short (≤3 days), mid (>3–5 days), or long (>5 days) precompetency periods (Fig. 2b; Table 1) and revealed notable taxa-specific patterns: most Merulinids, including the genera *Goniastrea*, *Dipsastraea* (formerly *Favia*), *Platygyra*, and *Oulophyllia* had 'short' precompetency periods while all 'long' precompetency-period species were of the genus *Acropora*. 'Mid' precompetency durations were the most common and included species from the greatest diversity of families (5). While these differences could potentially result in comparatively greater local retention in Merulinids and greater dispersal of Acroporids, it is difficult to determine whether this is reflected in population genetic structure for several reasons. Firstly, small spatial-scale genetic structure is rarely studied in corals[58,59], and these studies primarily investigate brooding species with extremely short pelagic larval durations (<1 day, i.e. refs. [60,61]), or compare brooding and broadcasting reproductive strategies (i.e. ref. [57]). Therefore, little data exist directly comparing genetic diversity

and dispersal capacity across space and taxa of broadcast spawners. Secondly, genetic structure within a local population can vary between recruits and adults, indicating that the adult populations sampled do not necessarily reflect larval connectivity[61]. Furthermore, broadcast spawning taxa with similar abilities to disperse can display different genetic structures (i.e. ref. [62]), with post-recruitment mortality processes also contributing to population structure. Despite these challenges, model-based and field-based data to date would suggest that recruitment through local retention may be common for several species and locations[8,10,20,41,63], but that enough propagules disperse long distances to ensure metapopulation connectivity across distant reefs[8,57,64,65]. Indeed, the modelled TC50 values were not significantly correlated with geographical species distributions, as estimated by Hughes et al.[66] (Fig. S1), suggesting that this variation in precompetency (potentially influenced by egg size) does not necessarily correlate with biogeographic distributions on evolutionary time scales; the lack of a correlation is likely due to a number of additional factors influencing connectivity including the degree of isolation, local population size, oceanography, inter- and intra-specific dynamics, and the competency window.

High within-cohort variation in precompetency is characteristic of spawning corals[2] and other invertebrates[21] and is influenced by many factors including the genetic diversity within the cohort[21] and the environmental conditions during larval rearing, which affect rates of larval development[9,67]. Our analysis applied a cohort-level threshold of 0.3 as the definition of 'competent' and thus the estimates here are most useful for understanding the onset of competency and, consequently, self-seeding dynamics[2]. On the other hand, the TC50 estimates modelled with a threshold of around 0.7 (Figs. 1b and 2b) would be most useful for understanding long-distance dispersal potential over evolutionary time scales[57]. While there was some variation in TC50 estimates amongst the thresholds used, the rank order of taxa was fairly consistent (Fig. 2b) suggesting that the gradual attainment of competency within the cohort occurs similarly amongst taxa.

Our study precluded a comprehensive investigation of the timing of loss of competency. In many species tested, however, larvae remained competent until the final available timepoint and some taxa demonstrated the potential for extended competency during the PLD (Fig. 3). These results corroborate the findings of Graham et al.[3,68], Harrison et al.[69], Wilson and Harrison[13], and others[11] who describe competency windows in excess of 100 days for some taxa (Table S1), and suggest that larval connectivity may extend to greater distances than initially estimated for some reef systems, such as those in Western Australia[63].

Interestingly, larval settlement fluctuated through time with peaks and troughs that were often consistent amongst cues within a species, and across taxa. This consistency suggests that fluctuations in settlement are likely reflective of larval behaviour and physiology and not cue variability alone, thereby challenging the assumption that competency gradually wanes over time. Periods of inactivity during the competency window, such as was seen in the bimodal pattern of every species tested beyond 40 days (*A. hyacinthus, D. matthaii, D. pallida, O. crispa,* and *Mycedium elephantotus*) (Fig. 3) could represent a 'bet-hedging strategy'[70], where a pulse of local settlement is followed by a metabolically inactive period of pelagic dispersal facilitating connectivity to non-natal areas for colonization when settlement competence increases once more. This is supported by evidence from Graham et al.[68] indicating that larvae can enter a state of low metabolic activity shortly after becoming competent to settle, supporting the capacity for long-distance dispersal. The 'desperate larval hypothesis'—the notion that larvae become less discriminatory as they age[21,36,71,72]—may also explain the resurgence in settlement behaviour at later timepoints; indeed, some species settled in

response to more cues during later timepoints (i.e. *D. matthaii* and *D. pallida*), while the prevalence of indiscriminate settlement—settlement in the absence of any cue—also increased through time (Fig. 3). However, the conditioning time of the disc necessarily increased with larval age and it's likely that benthic communities on all substrates changed in culture over time; thus, potential variability in inductivity of substrates cannot be ruled out as a driver of this resurgence. Whether this temporal variation has realized consequences for dispersal also depends heavily on survival throughout the pelagic period (Fig. 4).

Larvae were not fed in this study but there is mounting evidence that at least some species of coral larvae are capable of heterotrophic feeding[73] and the uptake of dissolved organic matter to supply amino acids[74]. Many larvae can also ingest symbionts through their mouth and incorporate them into their endoderm[75,76], and eggs of vertical transmitters and brooders host symbionts from parental colonies. Indeed, all vertical transmitters tested in this study (*P. cylindrica, P. lobata, M. digitata,* and *M. aequituberculata*) showed active settlement at the final time point (22–32 days) suggesting that nutrition from symbionts may aid in supporting long competency windows. Yet many horizontal transmitters also demonstrated extended competency in the absence of feeding. Whether heterotrophy and symbiont uptake during the larval stage can influence these durations, and therefore, metapopulation connectivity requires further investigation. Interestingly, Chamberland et al.[77] found that larvae of the brooding coral *Favia fragum* demonstrated increased swimming duration as symbiont density increased, but that symbiont-dense larvae were more thermally sensitive. Their results suggest context-dependent costs and benefits of larvae hosting symbionts; while well-provisioned larvae may be capable of longer-distance dispersal, they may be more sensitive to changing environmental conditions across the dispersed environment. How these costs and benefits influence connectivity in a changing climate warrants further study.

Conditioned reef rubble was the most effective settlement cue for 76% of species tested and was significantly better at inducing settlement than the CCA and biofilm disc in nearly all non-*Acropora* species. Less settlement on CCA and live biofilm discs compared with rubble fragments have several possible

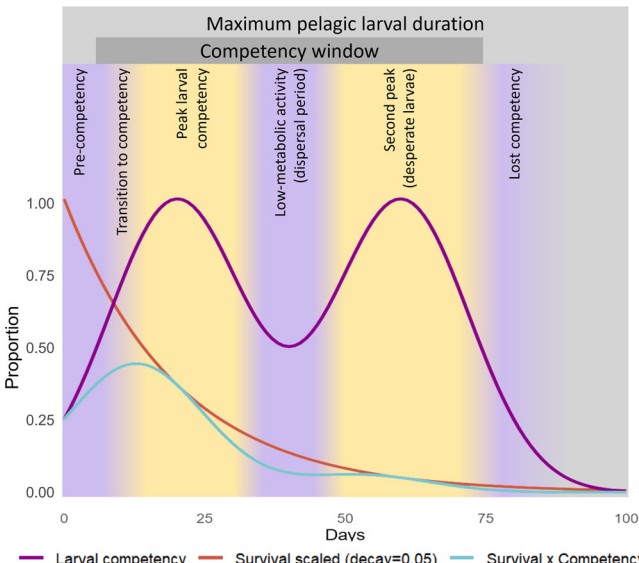

**Fig. 4 Conceptual model of settlement competency.** Conceptualization of a proposed bimodal pattern in larval competency overlaid on the maximum pelagic larval duration with a survival curve representing a decay rate of 0.05, and the product of competency and survival.

explanations that are not mutually exclusive. Firstly, the physical features of complex microhabitats within rubble may be an important consideration in a larva's settlement decision. Larvae are known to preferentially settle in microhabitats that offer refuge from external pressures[53,78–81] and to seek out habitats with lower light conditions[82]. Rubble fragments were texturally complex and offered ample microrefugia of various sizes that likely attracted larvae. Secondly, rubble fragments may have supported more complex and/or well-developed biofilms than the artificial discs, leading to the presence of stronger induction cues. Indeed, the presence of complex microhabitats likely created pockets of unique biofilm communities[83–85] that may have offered a wider diversity of potential inducers. It is also likely that biofilm discs supported early successional species (+4 weeks conditioned)[83] that were less mature and inductive than those on rubble fragments. Thirdly, while rubble was visually searched for CCA, it is likely that at least some fragments harboured cryptic and/or recruit CCA communities, which would be more diverse than the single CCA treatment tested and could be highly inductive[31–33,86]. As in Turnlund et al.[84], sequencing the cryptic taxa on more and less inductive fragments may provide further insights into key inductive and inhibitory communities. Finally, chemical extracts of some dead coral and CCA skeletons can induce coral larval settlement, indicating the potential presence of legacy inducers within these calcareous matrices and this may also be the case for the reef rubble applied in the current experiment. Regardless of the mechanism, reef rubble was overwhelmingly the best non-*Acropora* settlement cue, corroborating past studies[87,88] and highlighting the importance of further characterizing rubble substrates to identify potential inducers.

The best settlement inducer for Acroporidae varied by species and rubble was also a strong inducer across *Acropora*. This result was somewhat surprising because CCA is known to be a strong settlement inducers in *Acropora*[28,31–34,86,89]. However, Abdul Wahab et al.[33] recently demonstrated species-specific preferences amongst a broad taxonomic cross-section of coral/CCA pairings, and *Porolithon* was not the most universal CCA cue, corroborating previous findings[28,30,90,91]. Therefore, other CCAs may be as effective as rubble at inducing settlement across *Acropora* spp. but this remains untested.

The GLW-amide peptide Hym-248[92,93] significantly induced settlement in all *Acroporidae* species, but failed to induce settlement in nearly all non-*Acropora* species (Fig. 3; Supplementary Data 1; Table 2; Fig. S2), corroborating the findings of Erwin and Szmant[94] and indicating that this signalling pathway is likely not conserved amongst taxa within the Scleractinia, with high specificity of neuropeptide activators at low taxonomic levels. The testing of additional neuropeptides across a wider range of concentrations will improve our understanding of larval neurobiology.

While this laboratory-based experimental design was effective for assessing larval development and function, there are limitations to assessing larval behaviour in small-scale experiments. For example, larvae were confined in close proximity to physical and chemical cues for settlement and were not offered a choice. Thus, comparisons of absolute settlement success between cues should be interpreted cautiously. By contrast, the strategy of offering individual cues to identify the shortest precompetency period is an effective way of ensuring competency periods for larvae of diverse taxa are compared fairly. Indiscriminate larvae are more likely to settle soon after reaching competency, while selective species may spend longer in the plankton before encountering their preferred cue[21]. Because the preferred cue of most coral species is unknown, it is possible that we overestimated precompetency for those species that are selective (i.e., optimal inducers for settlement may not have been offered). Our data also indicated that while all but one species (*P. speciosa*) achieved

100% settlement in at least one replicate at some point during experimentation, the average peak competency wasn't at or near 100% for all taxa. The reasons for not reaching 100% competency could be related to many factors including species- or genotype-specific differences in maximum competence, sub-optimal larval health, or that none of the cues selected was optimal for that species[33]. More work is needed to understand this phenomenon. Furthermore, we tested single larval cohorts for most taxa (Table S2). Yet high within- and between-cohort variation in competency windows[2] has the potential to shift TC50 estimates. Thus, when applying these results to models of local retention and dispersal, it would be prudent to embed uncertainty around the estimates (Fig. 1a).

It wasn't possible to completely standardize the physical and biological settlement cues due to inherent variability in natural substrates and variations in spawning and settlement times. We attempted to account for this by: (i) using consistent (i.e., the same 'parent') rubble and CCA fragments across all treatments and species within a given month whenever possible; (ii) conditioning all substrates together, (iii) haphazardly loading substrates across all experimental wells to minimize bias, (iv) creating standard-sized substrates, and (v) applying robust replication. Despite this, there was variability in larval responses within treatments; whether this variability was related to changes in the quality or potency of the cue, or behavioural changes throughout ontogeny is difficult to untangle. Yet, overall patterns in settlement behaviour were remarkably consistent through time amongst treatments, and across species, supporting the patterns we describe.

Lastly, seawater temperature is known to influence survival and rate of embryogenesis[9,67,95,96]. Therefore, for consistency and to enable comparisons amongst taxa, larvae were cultured at 27–28 °C and maintained in a temperature-controlled environment during settlement. Consequently, experimental temperatures did not always precisely match those of natal reefs from which the broodstock originated. While these temperatures are generally considered non-stressful for GBR corals, it is unclear how slight divergence from the natal reef environment may have influenced larval development. Furthermore, the experimental conditions were not representative of the spawning temperatures across the entire geographic ranges of the taxa tested[66], and modelling has suggested that warming will decrease larval dispersal and connectivity[11]. Further testing is needed to understand how variations in temperature influence precompetency duration estimates and downstream dispersal.

Human activity and global climate change are fundamentally altering connectivity and recovery processes[11,97]. For example, declining adult densities can reduce larval supply directly[45], and indirectly via allee effects[50]. In addition, warmer temperatures can accelerate larval development[9,67,96], likely reducing the precompetency period and the maximum pelagic larval duration[98], leading to increased retention and reduced downstream dispersal and connectivity[11]. Figueiredo et al.[11] modelled the effect of 2 °C warming on coral larval dispersal and connectivity and identified significant decreases in distance travelled and number of connections amongst reefs, and an increase in local retention. The accompanied changes and shifts in ocean currents can also influence local retention and downstream dispersal of corals. At the reef scale, shifts in the benthic-community composition driven by marine heatwaves and acidifying conditions can impact the quality and quantity of larvae, their settlement cues, and the inducer-inhibitor ratio of organisms on the seafloor[99,100]. Yet the specificity for larval settlement cues[32,86,101], the duration of precompetency[12], and the PLD[3,13,102,103] vary considerably amongst species. Therefore, climate change is likely to unevenly affect the dispersal and recovery potential amongst species,

resulting in broad-scale and species-specific implications for biogeographic patterns in coral-community composition, the colonization of new habitats, and the expansion of species ranges under climate change.

There are four important management and restoration impacts resulting from this multi-species coral settlement study. The first is an appreciation of the variability within cohorts and between coral species. This variability may serve as a natural insurance policy against disturbances where local-retention and downstream connectivity are likely co-occurring and contributing to reef resilience. Second, the efficiency of seeding sexually produced corals for use in reef restoration purposes[50] can be improved through refined timing and prioritization of settlement by taxa based on our data, saving time and costs while maximizing output from aquaculture facilities. Third, the new knowledge of extended competency durations allows for the staged settlement and grow-out of spat prior to deployment, adding efficiencies and max-imising outputs of the coral production and restoration process. Finally, the multi-species data on precompetency and competency dynamics, and the potential impact on local retention and downstream connectivity will improve the spatial planning process[104–107] to help manage climate impacts and maximize the success of restoration efforts.

## Methods

**Coral collection and spawning**. Fecund colonies of 25 coral species were collected from locations along the GBR prior to the predicted 2017, 2018 and 2019 mass coral-spawning events in October and November each year (Table S2). Corals were iden-tified through visual characteristics of the live tissue and skeleton, with the aid of microscopy as required (i.e. for *Montipora* and *Porites* spp.), using Veron[108]. Corals were collected under permits G12/35236.1 and G19/43024.1 issued by the Great Barrier Reef Marine Park Authority (GBRMPA).

Corals were transported to the National Sea Simulator (SeaSim) at the Australian Institute of Marine Science (AIMS) via ship and maintained in flow-through outdoor aquaria under ambient light and at temperatures approximating those from source locations (based on daily 10-year average temperatures from in situ loggers or weather stations where available, or measured at the time of collection). On the predicted spawning nights[109], adult colonies were isolated and monitored for spawning activity. On the evenings of spawning, egg and sperm bundles or gametes were collected, bulk fertilized, and then transferred to larval rearing tanks (75 or 500 L), at a stocking density of ~0.5–1 larvae mL$^{-1}$. All spawning and culture metadata can be found in Table S2. Larval culture tanks received flow-through filtered (0.45 µm) seawater (FSW) at 27.0–28.0 °C and gentle aeration after ~16 h post-fertilization. Larval cultures were monitored and cleaned at least 3 times per day during the first two weeks and daily thereafter. Larval samples were haphazardly collected from homogenized bulk cultures immedi-ately prior to assay preparation. Therefore, larvae were cultured in the absence of settlement cues and maintained in the water column in culture until testing.

**Assay preparation**. Larval settlement assays were performed in sterile six-well cell-culture plates maintained in a constant tem-perature room (27–28 °C) under a 12:12 h light:dark cycle with ~20 µmol photons m$^{-2}$ s$^{-1}$, achieved with light emitting diode (LED) lights (AquaIllumination Sol White LEDs). Coral larvae (n = 10, nominally) were transferred by a 1.5 mL sterile plastic pipette into each well containing the cue to be tested, along with FSW, to a final volume of 10 mL. Settlement assays included six experimental treatments: (1) FSW, negative control; (2) live

*Porolithon cf. onkodes* fragment (~25 mm$^2$) (crustose coralline algae (hereafter 'Porolithon' or CCA)); (3) 5 µL of a 10% EtOH extract of *Porolithon*; (4) 10 mM concentration of GLW-amide peptide Hym-248; (5) a 12 mm diameter by 1 mm thick polylactic acid (PLA) 3D-printed disc (~113 mm$^2$) conditioned for +4 weeks in the SeaSim and hosting a biofilm (hereafter 'disc'); and (6) a fragment of coral skeletal rubble conditioned in SeaSim (~25 mm$^2$) covered in a live microbial biofilm with no visible CCA. Typically, each species, time and treatment combination was run with 6 replicate wells of 10 larvae each, although the number of larvae per replicate and the number of replicates per treatment occasionally varied due to larval availability (see Sup-plementary Code). Each assay was run with a new cohort of larvae. Each treatment is described in turn, below.

**Live *Porolithon cf. onkodes* fragments**. Fragments from the widely distributed and ecologically significant *Porolithon* species complex[110] were gathered during adult broodstock collection (~3–6 m depth) (under permit # G12/35236.1 issued by GBRMPA) using a hammer and chisel and were returned to the SeaSim and maintained in indoor aquaria under stable culture conditions to minimize changes to inductive potential over time. Each morning of assay set-up, bone cutters were used to cut small fragments (~25 mm$^2$) of CCA, each of which contained a live surface over a thin layer of cleanly cut calcium carbonate skeleton. Where possible, a continuous piece of CCA was cut and tested across all coral species on a given day and was tagged for use across multiple timepoints in a given month and year, to mini-mize variation in the CCA condition tested across assays. Only visibly healthy fragments with normal CCA colour and texture were used in assays.

**Porolithon cf. onkodes chemical extract**. Ethanolic extractions of *Porolithon* were generated following the protocol outlined in Whitman et al.[32]. Briefly, 25 mm$^2$ fragments of CCA were ground by mortar and pestle until 100 g of crushed material was obtained. 150 mL of absolute ethanol (EtOH; 100%) was added to the crushed material and mixed on a horizontal roller for 2 h at room temperature to form a CCA paste. The supernatant was then decanted and stored ($-20$ °C) to be used as the EtOH extract. An additional 150 mL of EtOH was added to the CCA paste for re-extraction and left mixing overnight to ensure all EtOH-soluble material was removed. The liquid EtOH extracts were combined, vacuum filtered (Whatman GF/F, 0.7 µm), and then prepared in 10% concentrations with EtOH (concentration equivalent to 0.03 g CCA mL$^{-1}$) and stored ($-20$ °C) until use in settlement assays. The extract was added to the treatment well and allowed to evaporate before the FSW and larvae were added (as per ref. [31]).

**GLW-amide peptide Hym-248**. The neurotransmitter GLW-amide peptide Hym-248 is known to induce metamorphosis and settlement in some *Acropora* species[92,94]; it was tested to deter-mine whether the neuropeptide signalling pathways that induce metamorphosis in *Acropora* are conserved across taxa. 100 µL of 1 mM concentrated peptide in 10 mL filtered seawater was used to achieve a final concentration of 10 µM per well. The peptide was added to the treatment well prior to the addition of FSW and larvae.

**Biofilm disc**. Microbial biofilms induce settlement in some coral species[83,111] and plastic substrates quickly recruit biofilm com-munities in aquaria. Thus, discs were developed to provide a standardized surface on which to colonize a biofilm. Polylactic acid (PLA) plastic discs (12 mm diameter, 1 mm thick) of a

neutral colour (brown) were 3D printed in sheets and placed horizontally and raised on polyvinyl chloride (PVC) racks, in semi-recirculating indoor aquaria (280 L, turnover 5 L min$^{-1}$, fluctuating light profile mimicking in situ conditions). After 1–3 months, the discs contained mature biofilm communities, and discs with similar conditioning duration and community composition (i.e. density of CCA, visible biofilm) were selected for assays. Discs were removed from the sheet using snips and added to FSW in the treatment wells using forceps before larvae were introduced.

**Conditioned reef rubble.** The conditioned reef rubble treatment (hereafter 'rubble') provided a structurally complex substrate harbouring a natural biofilm community known to induce settlement in some species[83,111,112] and offering microrefugia in which some larval taxa preferentially settle[53,78,79,113]. Fragments of rubble were sourced from the sumps of mature outdoor SeaSim aquaria. Only cylindrical rubble fragments originating from staghorn-type coral skeletons were used, to minimize variability

amongst fragments and for ease of generating uniform fragments. Rubble fragments (25–64 mm$^2$) were cut from parent fragments using bone cutters. Only fragments with a 'live' (outer) biofilm surface and without visible (by eye) CCA were used, although very small CCA recruits were likely present on some fragments. Furthermore, single large fragments were cut and tested across all assays on a given day and reserved for use across timepoints in a given month and year, to minimize variation in the rubble tested in the assays. When cut, some fragments revealed endolithic sponges, polychaetes and other bioeroding organisms; only 'clean' fragments without such organisms were tested. Clean rubble fragments were rinsed in FSW to remove any residue from cutting, and then placed in the centre of each treatment well with the outer surface facing up, prior to loading larvae into wells.

**Assay assessment.** All assays were scored 24–36 h after set-up by counting the number of larvae that had and had not demonstrated settlement under a standard dissection microscope. In species with fluorescent larvae, identification was aided by

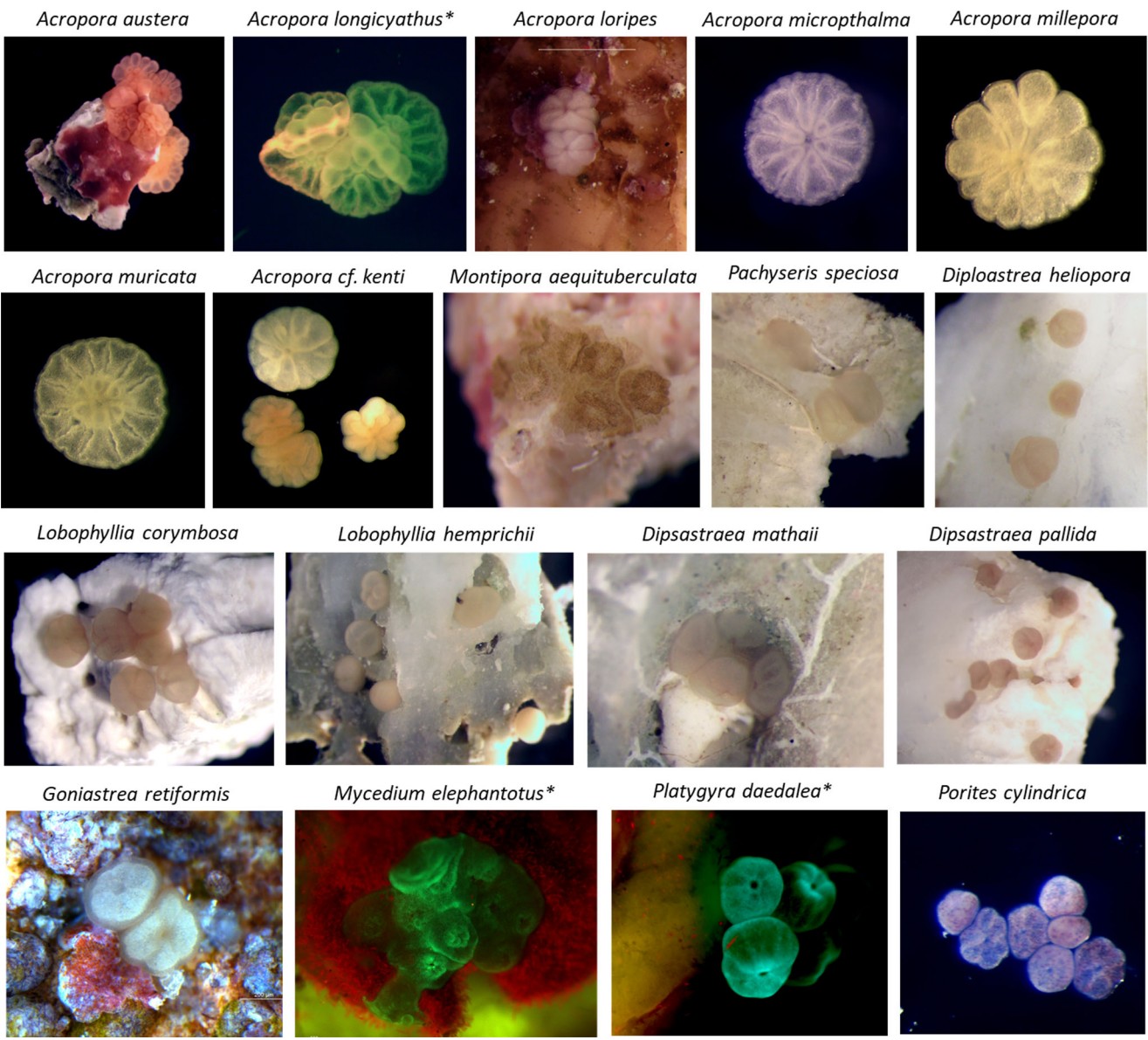

**Fig. 5 Recently settled coral spat.** Photomicrographs of coral spat of some species tested in the settlement assays, at ~24 h post settlement. * Indicates fluorescent (440–460 nm excitation) images demonstrating green fluorescent proteins in the coral tissue. Note that scales vary and are not included in most images; for approximate spat sizes, refer to Table 1 for oocyte diameter data.

applying a 440–460 nm fluorescent LED lighting system with a 500 nm long-pass emission filter (NIGHTSEA® Royal Blue) to the dissection microscope. Settlement was defined as the firm attachment of the larva to a surface, with a pronounced flattening of the oral–aboral axis and the presence of early septal mesenteries that radiated from a central mouth, with the deposition of a basal plate[86] (Fig. 5).

**Statistics and reproducibility: Precompetency period and competency duration**. To explore the duration of precompetency for each species, aggregated cohort-level settlement data were transformed into a binomial response as either competent (1) or not competent (0) to settle, based on the proportion of the larval cohort that settled in each replicate well. This threshold approach was used for three reasons: (i) to reduce potential biases that could arise from not presenting each species with its optimal inducer for settlement (i.e. ref. [33]), (ii) to enable the quantification of precompetency parameters (i.e. probability of time to competency), and (iii) to allow for cross-taxa comparisons. This approach also acknowledges cohort-level and species-level plasticity in individual behaviour and development (i.e. ref. [2]). A threshold settlement of 0.3 (30%) was used as a conservative proxy to define species-level competency. Above this threshold, the larvae within that replicate were considered to have demonstrated the potential for morphological and physiological settlement competency. In other words, 30% settlement was assumed to indicate that larvae were physiologically capable of settlement even if not all larvae chose to settle at that timepoint. Estimates of precompetency are also reported for a wide range of settlement threshold values (0.1–0.9) and the sensitivity of the analysis to this threshold is presented in the results.

The relationship between settlement and the interaction of larval age and settlement cue was then explored for each species using a hierarchical logistic model in a Bayesian framework[114]. The age at which the cohort was more likely to have reached larval competency than not, termed TC50 (i.e. Time to Competency at 50% probability; the inflection point at a probability of 0.5) was estimated for each species and settlement cue. To investigate the variability in TC50 amongst settlement cues, pairwise differences for each cue were calculated using exceedance probabilities.

All models included three No-U-Turn (Markov chain Monte Carlo (MCMC)) chains of 6000 iterations, thinned to a rate of 10, with a warm-up of 2000. MCMC mixture and convergence were assessed using trace plots, autocorrelation plots, Rhat and effective sample size diagnostics. Models were validated using DHARMa simulated residuals[115]. Models were run in R (R Core Team 2022) using the *brms* package[116,117] through *RStan*[118] and visualized using ggplot2[119].

Because the numbers of larvae in culture were limited for many species, a comprehensive study of competency window duration was not possible for all species. Assay assessments continued until larval supplies were exhausted, and this occurred as early as day 7 for *Pachyseris speciosa* and as late as day 77 for *Dipsastraea pallida*, *D. matthaii* and *Mycedium elephantotus*. Therefore, temporal patterns in settlement behaviour are described qualitatively by taxa.

**Statistics and reproducibility: Cue comparisons within species**. Raw settlement data were then used to quantify the preferential settlement amongst cues by species, and to investigate temporal patterns in settlement; a Bayesian generalized additive hierarchical model was used to represent the proportion of larvae settled against a cubic regression spline for larval age, conditional on settlement cue. Each model included a varying effect of well plate by age, to account for dependency structure. To protect against overfitting, spline knots were limited to a maximum of 5, which permitted a reasonable degree of trend tortuosity while preventing unreasonable perturbations from unusual values in the dataset. The area under each spline was calculated by numerical integration at a resolution of 100 for each posterior draw. Pairwise contrasts of the full posterior areas were compared using exceedance probabilities. Models were run and validated as described above.

**Egg diameter and TC50**. To investigate the relationship between egg size and TC50[8], average egg diameter data were collated for each species (Table 1) and modelled against TC50 ($P > 0.3$), using a generalized linear model with a Gaussian distribution, using the *stats* package in R (R Core Team 2022). All model diagnostics were assessed with the DHARMa package as described above.

**Reporting summary**. Further information on research design is available in the Nature Portfolio Reporting Summary linked to this article.

## Data availability
The permanent data record and raw data can be found here: https://apps.aims.gov.au/metadata/view/b5a63724-a339-488e-ae4d-b8a304371600.

## Code availability
Code is available at the github repository here: https://github.com/open-AIMS/larval_settlement_competency.

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

## Acknowledgements

We acknowledge the Bindal, Manbarra, Woppaburra, and Wulgurukaba Peoples as the Traditional Custodians of the Sea Countries where this research took place. The authors wish to acknowledge their Elder's past, present, and emerging, and their continuing spiritual connection to Sea Country. We thank J. Speaks, K. Allen, F. Brough, J. Grossman, and C. Sims for laboratory support and C. Kenkel, J. Gilmour and A. Heyward for valuable discussions. We thank the staff of the AIMS vessels and the National Sea Simulator for coral broodstock collection and spawning and larval husbandry support, respectively. This study was co-funded by The Reef Restoration and Adaptation Programme (RRAP), a partnership between the Australian Governments Reef Trust and the Great Barrier Reef Foundation, and the BHP–AIMS Australian Coral Reef Resilience Initiative (ACRRI).

## Author contributions

C.J.R. and A.P.N. designed the experiment. C.J.R., C.G., B.S., T.N.W. and C.A.P. conducted the experiments and collected the data. C.J.R. and M.L. analysed the data with input from E.A.T. and A.P.N. C.J.R. wrote the manuscript with contributions from all authors. All authors edited the manuscript and approved the final version.

## Competing interests

The authors declare no competing interests.
