## [Peer Review File · Communications Biology]

Larval precompetency and settlement behaviour in 25 Indo Pacific coral speciesReviewers' comments:

Reviewer #1 (Remarks to the Author):

Brilliant, extensive and very needed work from Randall et al. This manuscript describes the larval competency dynamics of 25 broadcast spawning corals and their settlement cues preferences. Aside from being new and important biological data for larval rearing for experimental and restoration purposes, it is essential to develop empirically-calibrated models for coral larval dispersal patterns that estimate reef connectivity and thus inform the design of effective networks of marine protected areas and optimal placement of restoration efforts. I am certain this will be a very well cited paper and thus I fully recommend it for publication. A few minor changes I would like the authors to address:

Line 81: I understand that the paper by Davies et al. described *Orbicella franksi* only settling after 20 days, but that is definitely not the pre-competency time for that species. It is well known that this species, when given the correct chemical cue, settles in 3-4 days (see for example Sneed et al 2014 <http://dx.doi.org/10.1098/rspb.2013.3086> which settled them on day 4). The authors likely did not have the correct settlement cue or had some inhibitory cue present unknown to them which prevented settlement earlier on. The settlement they observed on day 20 likely resulted from larvae getting desperate (like the second peak of settlement on some of the species you studied). Note that even just based on the egg size of *Orbicella franksi*, a 20 days pre-competency period would not make any sense. By the way, the authors of Davies et al. were advised about it by several reviewers on submissions they made to other journals (from which their paper was rejected) and decided to ignore it and publish it anyway on another journal, without even referring how it contradicts previous findings (Apologies for the vent!). In order not to perpetuate an inaccuracy, I ask you to please remove that reference. Thank you in advance!

Line 85: This paragraph would benefit from a small introduction (one or two lines at least) on the expected relationship between egg size and pre-competency period in corals, referring to Figueiredo et al. 2012 shows a relationship between egg size and larval motility.

Line 333: Can you report R^2 to determine what percent of the variance is explained by egg size alone?

Line 366 & 489: I suggest calling it instead "conditioned" Rubble because it is covered in microbial film (and likely even with some bits of CCA, zoox, and smell of adults?).

Line 431: and predation?

Citations in text: I apologize for recommending some of my own papers for citation, but several of them should not be overlooked. Figueiredo et al. 2022 describes long term (>4 months) survival and (>3 months) competency dynamics of *A. millepora* (in the appendix) and how these are altered by warming, presents a bio-physical dispersal model for the southern GBR under current and future/warming scenarios, and uses it to estimate source indices and simulate recovery from disturbances, and how these may be altered by climate change (lines 60 & 63 & 69 & 94 & 377-379 & 452). Figueiredo et al. 2012 reviews and describes lipid consumption during larval development (line 101). Bishop et al. 2006 is my favorite reference for the desperate larva hypothesis and other alternatives (line 118); see also references within. King et al. 2023 and Frys et al 2020 are good citations for how larval competency dynamics can be used to estimate connectivity and be used to decide where to restore (paragraph starting on 559). Chamberland et al 2017 studies the advantages and disadvantages of larvae having symbionts (lines 476-486).

King S., Saint-Amand A., Walker B.K., Hanert E., Figueiredo J. 2023 Larval dispersal patterns and connectivity of *Acropora* on Florida's Coral Reef and its implications for restoration. *Frontiers of Marine Science* 9:1038463. <http://doi.org/10.3389/fmars.2022.1038463>

Figueiredo J., Thomas C.J., Deleersnijder E., Lambrechts J., Baird A.H., Connolly S.R., Hanert E. 2022.

Global warming decreases connectivity among coral populations. *Nature Climate Change* 12: 83–87
<https://doi.org/10.1038/s41558-021-01248-7>

Figueiredo J., Baird A.H., Cohen M. F., Flot J.-F., Kamiki T., Meziane, T., Tsuchiya M., Yamasaki, H. 2012. Ontogenetic change in the lipid and fatty acid composition of scleractinian coral larvae. *Coral Reefs* 31:613–619
<http://dx.doi.org/10.1007/s00338-012-0874-3>

Bishop et al 2006 . <http://dx.doi:10.1093/icb/icd043>

Chamberland et al 2017 <http://dx.doi.org/10.1098/rspb.2017.0852>

Frys C., Saint-Amand A., Le Hénaff M., Figueiredo J., Kuba A., Walker B., Lambrechts J., Vallaey V., Vincent D., Hanert E. 2020. Fine-scale coral connectivity pathways in the Florida Reef Tract: Implications for conservation and restoration. *Frontiers of Marine Science* 7, 312.
<https://doi.org/10.3389/fmars.2020.00312>

Best regards,
Joana Figueiredo

Reviewer #2 (Remarks to the Author):

Randall et al. *Communications Biology*. Review

Overall Assessment

• Does the manuscript have technical or conceptual flaws that should prohibit its publication? If so, please provide details.

The manuscript does not have any major conceptual flaws. There are a few minor things that are detailed in the detailed feedback below.

Technically, the manuscript is generally robust. Common limitations that are associated with larval work and settlement cues are detailed over a couple of paragraphs in a dedicated section in the Discussion. There are some minor aspects in the M&M that need clarification, detailed in the detailed feedback below.

However, there is one aspect of the data compilation and analysis that I believe needs reworking or a much more thorough explanation in the M&M. As I understand it, the data have been aggregated and converted so that the response variable of settlement is based on a 30% threshold in each replicate well. This seems really strange, and I do not understand why the authors have chosen to do this rather than simply use the absolute settlement rates as done in all other studies. If I have understood correctly how the data have been aggregated, then I see a few issues with this (listed below).

1. The threshold value is arbitrarily chosen with no good justification as to why 30% was used. There are plots that show some differences that can occur among different aggregation levels (i.e., 2B, 3B) to try and evaluate how it changes or not, but these do not fully detail how things change (for example there is a 0.5 threshold in the response variable in 2B) and there are changes evident for many taxa at different aggregation thresholds.

2. The data are available to conduct the analyses on the absolute proportion settlement. This is the most robust way to analyse the data. It makes me wonder what may be happening with the data that may have influenced an aggregation approach for the analyses.

3. Utilising a threshold makes it difficult to compare with other studies (that all use absolute proportions). For example, in the discussion there is a direct comparison in pre-competency periods with a Figueiredo study, and there is a closely related pattern between those studies BUT a consistent offset is evident. My assumption is that the offset is related to the threshold approach used in this

study.

4. Conducting analysis on absolute proportion settlement may alter results / outcomes. For example, many of the competency curves show 100% settlement probability at peak competency periods, yet this is rarely (if ever?) realised in other studies. The reason that this is likely quite common in this study is because it is actually only 100% of the 6-wells had ≥ 3 out of 10 larvae within them settle. So, if we use an example of a consistent 3 out of 10 larvae settled in each of 6 replicates, that's actually only 0.3 settlement probability, but the way the data have been aggregated that would equal 1.0 settlement probability.

I suggest that the authors reanalyse the work using the absolute proportions rather than a threshold. If they decide that is too much work or there is good reason not to, I suggest that they provide a very good rationalisation as to why they have used a threshold, and then why they have chosen this threshold.

Although I do believe that I have understood the approach correctly, please forgive me if my understanding of the aggregation approach is incorrect. If that is the case, perhaps the aggregation approach needs more explanation in the M&M text around 268.

- Are the conclusions original? If not, please provide relevant references.

The conclusions by the authors are generally original. As described below, it's an unprecedented piece of work that greatly improves understanding of larval competency and PLD for a high diversity of common scleractinian corals. There are a few statements that need attention. In general, the authors have cited a broad range of studies, and I have suggested some additional studies that they may choose to include if they want to expand on some of the 'in prep' or single citation references in the text.

- Do you feel that the results presented are of immediate relevance for people in your own discipline or for a broader audience? If you recommend publication, please outline briefly what you consider to be the outstanding features.

Yes. The work conducted utilises a consistent approach with 21 scleractinian corals found on the Great Barrier Reef. Some of these larvae were held for up to ~75 days, and settlement assays run using multiple settlement cues. This level of diversity, consistency in assay, and time, adds greatly to what has been published thus far from across the world. The authors have done a great job in compiling those studies, to show the species and relevant pre-competency and maximum PLD from them. However, again, the data in this study are not directly comparable to those rates in that table given the aggregation threshold approach used in this study, so that needs to be addressed. Once addressed, I imagine that the data and findings from this study will be highly utilised, particularly for coral larvae connectivity studies occurring on the Great Barrier Reef, and I imagine will be highly cited by a broad audience.

- If you feel that specific additional experiments would strengthen the case for publication in Communications Biology, please provide suggestions.

Not required. Any potential 'additional/future work' is detailed in the manuscript discussion.

Detailed Feedback

Line 26: "to habitat selection" - Scale dependent - I suggest in context of the lab work in this study, it is 'micro-habitat'

Line 28: "ecologically significant" - How are they 'ecologically significant'?

Line 56-59: I think that this statement is a bit outdated (as shown in Table S1). There is definitely significant variability, but the major parameters of larval development time, precompetency, competency curves, based on zoox / azoox are known for a few model coral species - mainly some *Acropora* spp and merulinid spp

Line 60: "to changing environmental conditions" Add at least Figueiredo et al. 2014 Nat Cli Cha here

Line 69: "dispersal (population connectivity)" Add Gouezo et al. 2021 MEPS here

Line 74-75: "Rates of particle retention around reefs are often correlated with the rates at which competency is acquired" This isn't clear. Clarify that the particles are larvae and not just associated with the hydrodynamics of the reef such as a high residency time. It's a combination of both right - residence time of passive particle x competency period. Add citation.

Line 125-126: "Population connectivity and the retention of larvae drive local-scale demography and system-level metapopulation dynamics" Add Hock et al. 2018 Nat Comms

Line 158: "Table S2", but also relevant for species names being used throughout the document. Given the current updates to 'species' - particularly *Acropora* (e.g. Bridge et al. 2023) - I think the authors need to briefly describe how corals were identified in the current study and potentially use the 'cf' nomenclature as with the *Porolithon* cf *onkodes*.

Line 163-164: "flow-through outdoor aquaria at ambient light and temperature conditions matching source locations" Describe how light and temperature conditions matched to natal reefs for e.g., Palms and Keppels in a given spawning (e.g. Nov 2019)?

Line 169 and Line 179: "27.0-28.0°C" - Do these temperatures match natal reefs? And if not, maybe a slight discussion on that in the Limitations section is needed. i.e., it's better for comparison between taxa within this study, but may need to be used with caution in predicting natural dispersal if the temperatures differ.

Lines 173-174: "cultured in the absence of settlement cues and maintained in suspension in culture (i.e. simulating planktonic conditions)" Planktonic conditions include many particles that wouldn't be present in this system given 0.45 um filtering, so please reword that bit of the sentence 'simulating planktonic conditions'

Lines 197-199: "returned to the SeaSim and maintained in indoor aquaria under stable culture conditions to minimize changes to inductive potential over time" How did the change in light/flow conditions affect the CCA health, particularly when it had been kept for long periods of time in altered light/flow environments? These things can really affect the bioactivity of the cues from CCA.

Lines 235-236: "discs with similar conditioning duration and community composition (i.e. density of CCA, visible biofilm) were selected for assays" - Is that across a time period for any given species? A disc conditioned at 1 or 3 months would presumably have very different communities, so if the 1 month were given to a species when the larvae are younger and 3 month discs later when the larvae are older this could influence the settlement rates and inferences of competency optima.

Lines 242-243: "offering microrefugia in which some larval taxa preferentially settle" Also see Nozawa 2008 JEMBE, Doropoulos et al. 2016 Ecol Mono

Lines 248-249: Clarify whether 'visibility' was checked by eye or microscope? Many of the most inductive CCA are quite cryptic.

Lines 268-276: This seems really strange. Please explain why chosen to work at a replicate well level with an artificially defined 'competency threshold' of 0.3, when there is data for every individual? It's aggregating data when that is not necessary as a nested design can be utilised. And how/why is 0.3 the chosen threshold? It is an artificial threshold that could greatly change the outcomes of all of the results. For example, how would the plots and results of Fig. 2a, Fig. 4 differ if the threshold was set to 0.6, or 0.8, etc...? The work done in Fig. 2b and Fig 3b doesn't fully address this...

Lines 325-329: Really interesting results. However, I don't quite understand the rationale of categorising the continuous variable into those three groups when sometimes the difference is negligible (e.g., D mat vs O cri and P dae). The descriptions of shorter vs longer pre-competency times still holds, but rather than being binned they are continuous.

Line 333: very cool result.

Line 341-342: Be more explicit rather than 'many'. Looks like it's around 7 / 21 were roughly consistent for the best treatment. It'd be better to use a quantitative way of estimating whether there is consistency across thresholds within a species in the 'best' trt.

Line 343-344: And others, e.g., G fas, D mat, A glauca, A tenuis, P lobata, L hemprichii, have a 'notable' increases from 0.3 to >0.5. The difference between 0.3 and 0.7 is much clearer in Fig. 3B than 2B, so it may be easier for the reader to utilise 3B more and direct the reader to that.

Line 355-356: From the Figure, this appears driven by the 'disk', and thus relates to my comment in

the M&M about the age of the disks and the communities on them. Do disk age and larvae age covary?

Line 377-378: "These patterns challenge the long-held assumption that competency gradually wanes over time" That statement is not justified with the data presented. The data presented group things by a 30% threshold, so the temporal pattern of absolute competency of the total cohort is unknown. For example, the work by Connolly & Baird 2010 and Moneghetti et al. 2019, both use the proportion of all larvae rather than a threshold to derive their competency curves.

Also, although this is a bit harder to distinguish, for the corals in Fig 4 with curves longer than 40 days, the variability around the predicted fit becomes a lot higher on the right side of the bimodal curve. Is that because replication - temporal/# wells within a time/# larvae per well - were reduced at those later phases? Or, is it b/c there was more variability in the settlement thresholds between wells? When one larva metamorphoses in a well, it often stimulates the others to metamorphose. Or, is it otherwise?

Line 380: "identified a broadly effective settlement cue in reef rubble" See Golbuu & Richmond 2007 Mar Biol, Lee et al 2009 Cor Reefs

Lines 389-391: Could the slight offset be related to Figueiredo et al. using a proportion of the total rather than the proportion of a threshold (>0.3) in this study?

Lines 400-402: "Our results support this relationship and expand this to more taxa and reproductive strategies, indicating that larger embryos take longer to reach settlement competency." I suggest that the authors expand on why this may be. For example, could it be physiological and/or ecological and/or evolutionary mechanisms driving this relationship? Is there any evidence that big egg taxa are typically more widely distributed / small egg taxa more limited in distribution to support the association of egg size - competency - dispersal?

Lines 415-416: "small spatial-scale genetic structure is rarely studied in corals" Also see Foster et al. 2007 J Animal Ecol

Lines 424-425: "data to date would suggest that most recruitment is through local retention" Add citations for this because I'm not convinced this is demonstrated for spawning coral taxa. Most data from biophysical modelling studies suggests the opposite for well mixed systems such as the GBR, coastal NW WA, Caribbean, Palau.

Line 426-429: Neat comparison

Line 434-444: I appreciate this paragraph in the Discussion around using the threshold, and how using different thresholds may be more or less useful for making different types of predictions. However, it still doesn't justify why the authors don't use the absolute settlement rates in the first instance, rather than a threshold.

Lines 456-473: Very interesting finding and discussion points

Lines 496-498: It's likely that rubble fragments may have a high abundances of cryptic CCA, which are often highly inductive.

Lines 498-500: See Evenson et al. 2021 Ecology

Lines 506-507: This is unsubstantiated. Why is it important to do a chemical characterisation of rubble substrata to identify potential inducers?

Lines 512-513: This was also showed by Harrington et al. 2004, Price 2010, Doropoulos et al. 2012, Ritson-Williams et al. 2016, where Porolithon was much more limited in inductiveness compared to other CCA.

Lines 559-573: Solid paragraph

Line 586: "will improve the spatial planning process" See Doropoulos & Babcock 2018 Front Ecol Env Table S1: Also see Nozawa & Harrison 2005 Coral Reefs for *F. chinensis*, *G. aspera*, Doropoulos et al. 2018 Coral Reefs for *A. digitifera*, *A. millepora*, *G. retiformis*

Reviewer #1 (Remarks to the Author):

1. Brilliant, extensive and very needed work from Randall et al. This manuscript describes the larval competency dynamics of 25 broadcast spawning corals and their settlement cues preferences. Aside from being new and important biological data for larval rearing for experimental and restoration purposes, it is essential to develop empirically-calibrated models for coral larval dispersal patterns that estimate reef connectivity and thus inform the design of effective networks of marine protected areas and optimal placement of restoration efforts. I am certain this will be a very well cited paper and thus I fully recommend it for publication. A few minor changes I would like the authors to address.

Response: We sincerely thank the reviewer for the positive feedback and for the detailed review, which has helped us add important context and improve the manuscript. Please see below responses to the comments and changes requested.

Line 81: I understand that the paper by Davies et al. described *Orbicella franksi* only settling after 20 days, but that is definitely not the pre-competency time for that species. It is well known that this species, when given the correct chemical cue, settles in 3-4 days (see for example Sneed et al 2014 <http://dx.doi.org/10.1098/rspb.2013.3086> which settled them on day 4). The authors likely did not have the correct settlement cue or had some inhibitory cue present unknown to them which prevented settlement earlier on. The settlement they observed on day 20 likely resulted from larvae getting desperate (like the second peak of settlement on some of the species you studied). Note that even just based on the egg size of *Orbicella franksi*, a 20 days pre-competency period would not make any sense. By the way, the authors of Davies et al. were advised about it by several reviewers on submissions they made to other journals (from which their paper was rejected) and decided to ignore it and publish it anyway on another journal, without even referring how it contradicts previous findings (Apologies for the vent!). In order not to perpetuate an inaccuracy, I ask you to please remove that reference. Thank you in advance!

Response: We agree and have removed the Davies et al. reference. To illustrate the previously known range in coral precompetency period we replaced *O. franksi* with an Acroporid example.

2. Line 85: This paragraph would benefit from a small introduction (one or two lines at least) on the expected relationship between egg size and pre-competency period in corals, referring to Figueiredo et al. 2012 shows a relationship between egg size and larval motility.

Response: We agree. Introductory text was added to this paragraph at line 81: "Previous evidence also suggests that, for marine invertebrate larvae, propagule size may be an important predictor of precompetency duration^{8,20,21}. Yet whether this relationship is consistent across taxa and can be used to estimate such biological parameters requires further exploration."

3. Line 333: Can you report R^2 to determine what percent of the variance is explained by egg size alone?

Response: Yes, the adjusted R^2 (0.26) has been added to the text.

4. Line 366 & 489: I suggest calling it instead "conditioned" Rubble because it is covered in microbial film (and likely even with some bits of CCA, zoox, and smell of adults?).

Response: We agree and have edited the treatment name in the methods (line 240) and the lines suggested by the reviewer.

5. Line 431: and predation?

Response: We agree that predation as well as many other ecological relationships can also influence connectivity patterns and have edited the text to reflect this, capturing predation within the term 'inter- and intra-specific dynamics': "...the lack of a correlation is likely due to a number of additional factors influencing connectivity including the degree of isolation, local population size, oceanography, inter- and intra-specific dynamics, and the competency window."

Citations in text: I apologize for recommending some of my own papers for citation, but several of them should not be overlooked. Figueiredo et al. 2022 describes long term (>4 months) survival and (>3 months) competency dynamics of *A. millepora* (in the appendix) and how these are altered by warming, presents a biophysical dispersal model for the southern GBR under current and future/warming scenarios, and uses it to

estimate source indices and simulate recovery from disturbances, and how these may be altered by climate change (lines 60 & 63 & 69 & 94 & 377-379 & 452). Figueiredo et al. 2012 reviews and describes lipid consumption during larval development (line 101). Bishop et al. 2006 is my favorite reference for the desperate larva hypothesis and other alternatives (line 118); see also references within. King et al. 2023 and Frys et al 2020 are good citations for how larval competency dynamics can be used to estimate connectivity and be used to decide where to restore (paragraph starting on 559). Chamberland et al 2017 studies the advantages and disadvantages of larvae having symbionts (lines 476-486).

Response: We thank the reviewer for pointing us to these relevant and important references. They have been reviewed and added throughout the text, where appropriate.

Reviewer #2 (Remarks to the Author):

6. *Does the manuscript have technical or conceptual flaws that should prohibit its publication? If so, please provide details?* The manuscript does not have any major conceptual flaws. There are a few minor things that are detailed in the detailed feedback below. Technically, the manuscript is generally robust. Common limitations that are associated with larval work and settlement cues are detailed over a couple of paragraphs in a dedicated section in the Discussion. There are some minor aspects in the M&M that need clarification, detailed in the detailed feedback below.

Response: We sincerely thank the reviewer for their thorough evaluation and the positive feedback on our manuscript. Please see below for responses to each point raised.

7. There is one aspect of the data compilation and analysis that I believe needs reworking or a much more thorough explanation in the M&M. As I understand it, the data have been aggregated and converted so that the response variable of settlement is based on a 30% threshold in each replicate well. This seems really strange, and I do not understand why the authors have chosen to do this rather than simply use the absolute settlement rates as done in all other studies. If I have understood correctly how the data have been aggregated, then I see a few issues with this (listed below).

Response: We recognize that this analysis deviates from convention, but we chose to use this threshold approach for several reasons, and thus have opted to add further justification to the text. Below we summarize the main reasons for using the threshold approach, justify why we used a 30% threshold and describe how the approach and the results correspond to previous literature. We also acknowledge that the justification and analysis objectives were unclear in the original text and have thus edited the manuscript for clarity. This is a general response to the reviewer comment and in subsequent comments we point to specific changes in the text.

Why use a threshold approach?

- **To reduce unknown biases and increase comparability amongst taxa.** It is likely that there are unknown biases that arose from possibly not presenting each species with its optimal inducer for settlement (i.e. Abdul Wahab et al. 2023). This is also why the time to competency analysis was modelled using the most successful inducer for each species rather than averaging across effective inducer treatments. Thus, using the threshold approach enables more reliable comparisons amongst species than would metrics that are based on absolute settlement success, which can be affected more by experimental artefacts related to the settlement cues presented.
- **To define a species-level estimate of the pre-competency parameter.** Taking the mindset that larvae are either physiologically capable of settlement (1) or they are not (0), we decided to conceptually ‘transform’ the data from settlement rates per se (the focus of earlier literature), to a metric more consistent with competency. For clarity around communicating results and implications, and to give direct pathways for including these outcomes in models (and management), we used this binary model of competency, which requires the threshold decision. We clearly acknowledged that there is plasticity and variability around this physiological switch, and presented the implications of this variability. To further increase transparency, we have added a supplementary table with all treatment by species by time average settlement rates.
- **Using absolute settlement rate is restrictive.** Asking nuanced questions like precompetency duration and the time to 50% probability of cohort competency are not possible to model using absolute settlement rate. Here we are less interested in settlement rates and more interested in these species-specific parameters. We

acknowledge that these objectives were not clearly explained in the original text and we have edited to clarify this in the revision.

- **In summary, consider this threshold approach is less biased, more conservative in estimating precompetency, and is presented transparently. Importantly, we also note that for the temporal analysis done using GAMs, proportional data were used (Figure 4), and the text was edited to clarify this.**

Why use a 30% threshold?

- 30% was chosen because we contend that if at least 30% of the cohort is capable of and demonstrates settlement, then it would indicate that the larvae *can* reach competency by this time. Not all larvae will settle immediately upon reaching competency, and not all larvae will reach competency at the same time (i.e. there is plasticity in this metric, also acknowledged in the text). Thus, we considered 30% a justifiable threshold.
- However, we also agree that other thresholds may be more suitable for estimating this metric depending on its use, and therefore we presented the model estimates for every threshold from 0.1 to 0.9 in 0.1 increments. It is left up to the reader to choose which values are most appropriate for their application.
- Finally, based on Figure 2B, the 0.3 threshold appears to form the first plateau in the TC50 estimate data, suggesting that beyond 0.3, the TC50 estimates stabilise, until very high thresholds are reached.
- **In short, we think that this 30% threshold is conservative, ensures repeatability and we are fully transparent with the presentation of all thresholds, and indeed the results are robust across a wide range of thresholds (Fig 2B).**
- Regarding the '0.5 threshold in the response variable', it is likely this reviewer mistook our calculated TC50 (the modelled inflection point at 50% *probability* of competency), with 50% settlement. We have also clarified this in the text.

How does this compare with past studies?

- As described above, we are less interested in settlement rates per se and more interested in species-specific competency parameters. But, to aid comparisons with past studies, we have added a supplementary table with all treatment by species by time average settlement rates. Furthermore, the GAMs (Figure 4) were modelled using settlement rate data, as suggested by the reviewer, and are useful for making more direct comparisons with past studies. The GAMs were presented to address a different question (cue comparison) rather than time to competency.
- Some past studies (i.e. Figueredo et al. 2013) do use a binomial model of settlement and can be more easily compared with our results.

We also note that all raw data are presented in supplementary files and the html code and plotted in supplementary figures.

8. 1. The threshold value is arbitrarily chosen with no good justification as to why 30% was used. There are plots that show some differences that can occur among different aggregation levels (i.e., 2B, 3B) to try and evaluate how it changes or not, but these do not fully detail how things change (for example there is a 0.5 threshold in the response variable in 2B) and there are changes evident for many taxa at different aggregation thresholds.

Response: Please see above. In addition, a justification of 30% was added to the manuscript: "A threshold settlement of 0.3 (30%) was used as a conservative proxy to define species-level competency. Above this threshold, the larvae within that replicate were considered to have demonstrated the potential for morphological and physiological settlement competency. In other words, 30% settlement was assumed to indicate that larvae were physiologically capable of settlement even if not all larvae chose to settle at that timepoint. Estimates of precompetency are also reported for a wide range of settlement threshold values (0.1-0.9) and the sensitivity of the analysis to this threshold is presented in the results."

9. 2. The data are available to conduct the analyses on the absolute proportion settlement. This is the most robust way to analyse the data. It makes me wonder what may be happening with the data that may have influenced an aggregation approach for the analyses.

Response: We contend that the threshold approach is indeed very robust and less biased, allowing for better inter-specific comparisons. Please see above. We also note that the raw data are available in the supplementary, plotted in the code file, and we added a supplementary table of average settlement data by species, time and treatment.

10. 3. Utilising a threshold makes it difficult to compare with other studies (that all use absolute proportions). For example, in the discussion there is a direct comparison in pre-competency periods with a Figuerido study, and there is a closely related pattern between those studies BUT a consistent offset is evident. My assumption is that the offset is related to the threshold approach used in this study.
Response: We think it's unlikely that the offset is related to the threshold approach. If transforming our data to binary values resulted in underestimating time to competency, our estimates would be less than those by Figuerido et al. Yet in all 3 cases, they are slightly greater (although within the credible interval estimates). We suggest that these differences are instead likely due to statistical differences across studies, and possibly cohort-level effects. Furthermore, Figuerido et al. 2013 also uses a binomial model of settlement, with tweaks/constraints to 'improve the statistical properties.'
11. 4. Conducting analysis on absolute proportion settlement may alter results / outcomes. For example, many of the competency curves show 100% settlement probability at peak competency periods, yet this is rarely (if ever?) realised in other studies. The reason that this is likely quite common in this study is because it is actually only 100% of the 6-wells had ≥ 3 out of 10 larvae within them settle. So, if we use an example of a consistent 3 out of 10 larvae settled in each of 6 replicates, that's actually only 0.3 settlement probability, but the way the data have been aggregated that would equal 1.0 settlement probability.
Response: Firstly, we believe this misunderstanding may stem from our figure labels, which have been changed to clarify: Figure 2A y-axis label was changed from 'Cohort settlement probability ($P > 0.3$)' to 'Cohort settlement competency probability ($P > 0.3$)' and Figure 2B y-axis label from 'Days to > 0.5 settlement probability' to 'Days to > 0.5 probability of settlement competency'. We also edited the methods text to clarify that TC50 is 50% probability of competency (defined as settlement $> 30\%$) (not 50% settlement). Secondly, because we are quantifying competency probability and not settlement %, our curve should indeed reach 100%. In addition, absolute settlement did reach 100% for every species but one (*Pachyseris speciosa*, which peaked at 90%), which is why estimated TC50 values for thresholds from 0.1 to 0.9 are available for all taxa for at least one settlement cue. Also see edited text in the first paragraph of the limitations section.
12. I suggest that the authors reanalyse the work using the absolute proportions rather than a threshold. If they decide that is too much work or there is good reason not to, I suggest that they provide a very good rationalisation as to why they have used a threshold, and then why they have chosen this threshold. Although I do believe that I have understood the approach correctly, please forgive me if my understanding of the aggregation approach is incorrect. If that is the case, perhaps the aggregation approach needs more explanation in the M&M text around 268.
Response: Please see responses above for a rationalisation. We respectfully opted to add a justification to the text rather than re-analyse the data.
13. *Are the conclusions original?* If not, please provide relevant references? The conclusions by the authors are generally original. As described below, it's an unprecedented piece of work that greatly improves understanding of larval competency and PLD for a high diversity of common scleractinian corals. There are a few statements that need attention. In general, the authors have cited a broad range of studies, and I have suggested some additional studies that they may choose to include if they want to expand on some of the 'in prep' or single citation references in the text.
Response: Thank you for the suggestions, which have been actioned. Please see responses below.
14. *Do you feel that the results presented are of immediate relevance for people in your own discipline or for a broader audience? If you recommend publication, please outline briefly what you consider to be the outstanding features.* Yes. The work conducted utilises a consistent approach with 21 scleractinian corals found on the Great Barrier Reef. Some of these larvae were held for up to ~ 75 days, and settlement assays run using multiple settlement cues. This level of diversity, consistency in assay, and time, adds greatly to what has been published thus far from across the world. The authors have done a great job in compiling those studies, to show the species and relevant pre-competency and maximum PLD from them. However, again, the data in this study are not directly comparable to those rates in that table given the aggregation threshold approach used in this study, so that needs to be addressed. Once addressed, I imagine that the data and findings from this study will be

highly utilised, particularly for coral larvae connectivity studies occurring on the Great Barrier Reef, and I imagine will be highly cited by a broad audience.

Response: We appreciate the reviewer's positive feedback. Please see responses above regarding the analysis approach and comparability with past studies.

15. *If you feel that specific additional experiments would strengthen the case for publication in Communications Biology, please provide suggestions. Not required. Any potential 'additional/future work' is detailed in the manuscript discussion.*
16. Line 26: "to habitat selection" - Scale dependent - I suggest in context of the lab work in this study, it is 'micro-habitat'
Response: We agree that microhabitat is the appropriate spatial scale tested in this study (i.e. line 493). However, the sentence referred to in the abstract is of a more general nature and is intended to highlight the fact that settlement cues impact habitat selection, patterns of which can occur across multiple spatial scales. Thus, we have opted to leave the abstract text unchanged.
17. Line 28: "ecologically significant" - How are they 'ecologically significant'?
Response: These taxa were selected based on long-term monitoring data (AIMS LTMP) of corals across the GBR that indicate the dominant and/or abundant genera across reefs. We have clarified the text here to state 'abundant and widely distributed'. Given that the statement is in the abstract, we were unable to include a citation to the latest LTMP report.
18. Line 56-59: I think that this statement is a bit outdated (as shown in Table S1). There is definitely significant variability, but the major parameters of larval development time, precompetency, competency curves, based on zoox / azoox are known for a few model coral species - mainly some Acropora spp and merulinid spp
Response: We agree and have edited the statement to focus on the lack of data across taxa: "biological data are lacking for key coral life-history parameters that are known to shape connectivity, for the vast majority of species."
19. Line 60: "to changing environmental conditions" Add at least Figueiredo et al. 2014 Nat Cli Cha here
Response: Citation to Figueiredo et al. 2014 and 2022 added here.
20. Line 69: "dispersal (population connectivity)" Add Gouezo et al. 2021 MEPS here
Response: Citation added.
21. Line 74-75: "Rates of particle retention around reefs are often correlated with the rates at which competency is acquired" This isn't clear. Clarify that the particles are larvae and not just associated with the hydrodynamics of the reef such as a high residency time. It's a combination of both right - residence time of passive particle x competency period. Add citation.
Response: The text was edited to clarify, and citations were added: ". Indeed, rates of larval retention (modelled as particles) around reefs are often a similar order to the rates at which competency is acquired^{8,20}. Cetina-Heredia and Connolly²⁰ modelled larval retention times for a variety of reef types using a hydrodynamic model and found that mean water residence times were 0.5 to 5.6 days. Small differences in the precompetency period, therefore, can have significant consequences for local retention and dispersal."
22. Line 125-126: "Population connectivity and the retention of larvae drive local-scale demography and system-level metapopulation dynamics" Add Hock et al. 2018 Nat Comms
Response: Citation added.
23. Line 158: "Table S2", but also relevant for species names being used throughout the document. Given the current updates to 'species' - particularly Acropora (e.g. Bridge et al. 2023) - I think the authors need to briefly describe how corals were identified in the current study and potentially use the 'cf' nomenclature as with the Porolithon cf onkodes.

Response: We agree. Text has been added to describe the identification method, and *Acropora tenuis* has been changed to *Acropora cf. kenti* (*tenuis*) with a citation to Bridge et al. 2023, throughout: “Corals were identified through visual characteristics of the live tissue and skeleton, with the aid of microscopy as required (i.e. for *Montipora* and *Porites* spp.), using Veron ⁵⁵.”

24. Line 163-164: “flow-through outdoor aquaria at ambient light and temperature conditions matching source locations” Describe how light and temperature conditions matched to natal reefs for e.g., Palms and Keppels in a given spawning (e.g. Nov 2019)?

Response: Text edited: “Corals were transported to the National Sea Simulator (SeaSim) at the Australian Institute of Marine Science (AIMS) via ship and maintained in flow-through outdoor aquaria under ambient light and at temperatures approximating those from source locations (based on daily 10-year average temperatures from *in situ* loggers, or weather stations where available or measured at the time of collection).”

25. Line 169 and Line 179: “27.0-28.0°C” - Do these temperatures match natal reefs? And if not, maybe a slight discussion on that in the Limitations section is needed. i.e., it’s better for comparison between taxa within this study, but may need to be used with caution in predicting natural dispersal if the temperatures differ.

Response: As suggested, we maintained a consistent experimental temperature to enable comparisons amongst taxa, although the experimental conditions were generally within 2 degrees of natal reefs. We added a short paragraph to the limitation section as suggested: “Lastly, seawater temperature is known to influence survival and rate of embryogenesis ^{9,82,105,106}. Therefore, for consistency and to enable comparisons amongst taxa, larvae were cultured at 27-28 °C and maintained in a temperature-controlled environment during settlement. Consequently, experimental temperatures did not always precisely match those of natal reefs from which the broodstock originated. While these temperatures are generally considered non-stressful for GBR corals, it is unclear how slight divergence from the natal reef environment may have influenced larval development. Furthermore, the experimental conditions were not representative of the spawning temperatures across the entire geographic ranges of the taxa tested ⁸¹, and modelling has suggested that warming will decrease larval dispersal and connectivity ¹¹. Further testing is needed to understand how variations in temperature influence precompetency duration estimates and downstream dispersal.”

26. Lines 173-174: “cultured in the absence of settlement cues and maintained in suspension in culture (i.e. simulating planktonic conditions)” Planktonic conditions include many particles that wouldn't be present in this system given 0.45 um filtering, so please reword that bit of the sentence 'simulating planktonic conditions'

Response: We agree, and have edited the text to: “Therefore, larvae were cultured in the absence of settlement cues and maintained in suspension the water column until testing.”

27. Lines 197-199: “returned to the SeaSim and maintained in indoor aquaria under stable culture conditions to minimize changes to inductive potential over time” How did the change in light/flow conditions affect the CCA health, particularly when it had been kept for long periods of time in altered light/flow environments? These things can really affect the bioactivity of the cues from CCA.

Response: We can't say if/how the bioactivity of the CCA change through time as to do so would require an additional large lab experiment and was outside the scope of this study. However, only apparently normal/healthy and growing CCA were used in the assays. Given how inductive the CCA was generally, and that the CCA fragments were used over many months to induce larvae from multiple spawnings (i.e. as competency was waning in Nov spawners it was ramping up for Dec spawners, using the same CCA fragments), we suggest that our criteria of ‘only visibly healthy fragments with normal CCA colour and texture were used in assays’ was sufficient. Furthermore, we did not assume CCA was the most optimal inducer for all timepoints/species, with time to competency calculations instead based on the best response to any of the four inducers. No change made.

28. Lines 235-236: “discs with similar conditioning duration and community composition (i.e. density of CCA, visible biofilm) were selected for assays” - Is that across a time period for any given species? A disc conditioned at 1 or 3 months would presumably have very different communities, so if the 1 month were given to a species

when the larvae are younger and 3 month discs later when the larvae are older this could influence the settlement rates and inferences of competency optima.

Response: Yes, across a time period for a given species. Also, see Response 35 below.

29. Lines 242-243: “offering microrefugia in which some larval taxa preferentially settle” Also see Nozawa 2008 JEMBE, Doropoulos et al. 2016 Ecol Mono

Response: Citations added.

30. Lines 248-249: Clarify whether ‘visibility’ was checked by eye or microscope? Many of the most inductive CCA are quite cryptic.

Response: The text was edited to clarify: “Only fragments with a ‘live’ (outer) biofilm surface and without visible (by eye) CCA were used, although very small CCA recruits were likely present on some fragments.” We also added commentary about this in the discussion: “Thirdly, while rubble was visually searched for CCA, it is likely that at least some fragments harboured cryptic and/or recruit CCA communities, which would be more diverse than the single CCA treatment tested, and could be highly inductive ^{31–33,65}.”

31. Lines 268-276: This seems really strange. Please explain why chosen to work at a replicate well level with an artificially defined 'competency threshold' of 0.3, when there is data for every individual? It's aggregating data when that is not necessary as a nested design can be utilised. And how/why is 0.3 the chosen threshold? It is an artificial threshold that could greatly change the outcomes of all of the results. For example, how would the plots and results of Fig. 2a, Fig. 4 differ if the threshold was set to 0.6, or 0.8, etc...? The work done in Fig. 2b and Fig 3b doesn't fully address this...

Response: Please see responses above for an explanation of the approach and 0.3 threshold decision. Additionally, we note that the GAM analyses did indeed use proportional data for settlement (please see responses above for more details).

Lines 325-329: Really interesting results. However, I don't quite understand the rationale of categorising the continuous variable into those three groups when sometimes the difference is negligible (e.g., D mat vs O cri and P dae). The descriptions of shorter vs longer pre-competency times still holds, but rather than being binned they are continuous.

Response: Precompetency duration is a continuous variable but we chose to categorize them for ease of making broad taxonomic comparisons and identifying patterns (Table 1). We envision that the median precompetency duration for each category could be used in future connectivity modelling, thus providing insight into dynamics for short, mid and long precompetency taxa. However, we have clearly and transparently provided the TC50 estimates and credible intervals for every species so the continuous data are also available.

32. Line 333: very cool result.

Response: Thanks.

33. Line 341-342: Be more explicit rather than 'many'. Looks like it's around 7 / 21 were roughly consistent for the best treatment. It'd be better to use a quantitative way of estimating whether there is consistency across thresholds within a species in the 'best' trt.

Response: We understand the Reviewer's desire for this to be quantified. However, because this is open to interpretation and depends on how one defines 'consistent across much of the range in thresholds' it is not straightforward to provide this. Instead, we refer the reader to Figure 2B to make that determination, and have corrected the figure citation in the text.

34. Line 343-344: And others, e.g., G fas, D mat, A glauca, A tenuis, P lobata, L hemprichii, have a 'notable' increases from 0.3 to >0.5. The difference between 0.3 and 0.7 is much clearer in Fig. 3B than 2B, so it may be easier for the reader to utilise 3B more and direct the reader to that.

Response: We agree and have added a reference to figure 3B here.

35. Line 355-356: From the Figure, this appears driven by the 'disk', and thus relates to my comment in the M&M about the age of the disks and the communities on them. Do disk age and larvae age covary?
Response: In most cases, the resurgence in settlement seems to be driven by rubble (teal colour) but we take the Reviewer's point and have edited the following text: "The 'desperate larval hypothesis'—the notion that larvae become less discriminatory as they age^{21,36,86,87}—may also explain the resurgence in settlement behaviour at later timepoints; indeed, some species settled in response to more cues during later timepoints (i.e. *D. matthaii* and *D. pallida*), while the prevalence of indiscriminate settlement—settlement in the absence of any cue—also increased through time (Figure 4). However, the conditioning time of the disc necessarily increased with larval age and it's likely that benthic communities on all substrates changed in culture over time; thus, potential variability in inductivity of substrates cannot be ruled out as a driver of this resurgence. Whether this temporal variation has realized consequences for dispersal also depends heavily on survival throughout the pelagic period (Figure 5)."
36. Line 377-378: "These patterns challenge the long-held assumption that competency gradually wanes over time" That statement is not justified with the data presented. The data presented group things by a 30% threshold, so the temporal pattern of absolute competency of the total cohort is unknown. For example, the work by Connolly & Baird 2010 and Moneghetti et al. 2019, both use the proportion of all larvae rather than a threshold to derive their competency curves. Also, although this is a bit harder to distinguish, for the corals in Fig 4 with curves longer than 40 days, the variability around the predicted fit becomes a lot higher on the right side of the bimodal curve. Is that because replication - temporal/# wells within a time/# larvae per well - were reduced at those later phases? Or, is it b/c there was more variability in the settlement thresholds between wells? When one larva metamorphoses in a well, it often stimulates the others to metamorphose. Or, is it otherwise?
Response: We recognise the possible confusion regarding 1) the analysis of time to competency, which uses the threshold approach (figure 2), and 2) the GAMs, which are used to model settlement over time (Figure 4). Indeed, the GAMs use the 'raw' settlement data (i.e., the number settled and the number not settled per replicate) and the temporal trends in settlement are discussed using these models. Therefore, we stand by the conclusion that competency wanes over time. However, we recognize that this was unclear and thus edited the text in the methods: "Raw data were then used to quantify the preferential settlement amongst cues by species, and to investigate the temporal patterns in settlement; a Bayesian generalized additive hierarchical model was used to represent the proportion of larvae settled against a cubic regression spline for larval age, conditional on settlement cue." Note that we state the proportion of larvae settled in these methods.
37. Line 380: "identified a broadly effective settlement cue in reef rubble" See Golbuu & Richmond 2007 Mar Biol, Lee et al 2009 Cor Reefs
Response: References were added to the discussion section: "Inter- and intra-specific patterns in response to settlement cues".
38. Lines 389-391: Could the slight offset be related to Figueiredo et al. using a proportion of the total rather than the proportion of a threshold (>0.3) in this study?
Response: We do not believe this offset is due to the threshold method applied in the analysis. Please see response #10 above for justification.
39. Lines 400-402: "Our results support this relationship and expand this to more taxa and reproductive strategies, indicating that larger embryos take longer to reach settlement competency." I suggest that the authors expand on why this may be. For example, could it be physiological and/or ecological and/or evolutionary mechanisms driving this relationship? Is there any evidence that big egg taxa are typically more widely distributed / small egg taxa more limited in distribution to support the association of egg size - competency - dispersal?
Response: We have included additional information on "why" egg size may be linked to pre-competency windows: "The rate of development through ontogeny is often size dependent, influenced by the average amount of energy needed to create cells (i.e. more rapid development for cells of less mass) (Gillooly et al., 2002). Egg size governs larval size, with smaller conspecific marine larvae invertebrates reaching competency earlier (Nanninga and Berumen, 2014). In corals, egg size has been correlated with time to motility among

species⁸ and propagule size has been positively correlated with precompetency duration in brooding corals⁷³.”

There is no direct evidence that corals with large eggs are more widely distributed. However, the following paragraph discussed the links between taxa-specific competency and dispersal and the lack of evidence that dispersal is the primary driver of observed community composition. To avoid repetition, we linked these sections by specifying that variation in precompetency is potentially influenced by egg size. Any further discussion on the potential physiological or evolutionary links between egg size and community distribution would be speculative and beyond the scope of the study.

40. Lines 415-416: “small spatial-scale genetic structure is rarely studied in corals” Also see Foster et al. 2007 J Animal Ecol
Response: Citation added.
41. Lines 424-425: “data to date would suggest that most recruitment is through local retention” Add citations for this because I'm not convinced this is demonstrated for spawning coral taxa. Most data from biophysical modelling studies suggests the opposite for well mixed systems such as the GBR, coastal NW WA, Caribbean, Palau.
Response: Thank you for this comment. Recruitment and local retention is confounded in the literature and results are both species-specific and context (seascape) specific. We have softened the language to reflect this and added references demonstrating these dynamics across taxa. See edited text in the section “Taxa-specific patterns and their implications for community composition”. Refs: Figueiredo et al 2013; Cetina-Heredia & Connolly 2011; Grimaldi et al. 2022; Gilmour et al. 2009
42. Line 426-429: Neat comparison
Response: Thanks!
43. Line 434-444: I appreciate this paragraph in the Discussion around using the threshold, and how using different thresholds may be more or less useful for making different types of predictions. However, it still doesn't justify why the authors don't use the absolute settlement rates in the first instance, rather than a threshold.
Response: Please see responses above.
44. Lines 456-473: Very interesting finding and discussion points
Response: Thanks!
45. Lines 496-498: It's likely that rubble fragments may have a high abundances of cryptic CCA, which are often highly inductive.
Response: We agree and have added the following sentence to the discussion: “Thirdly, while rubble was visually searched for CCA, it is likely that at least some fragments harboured cryptic and/or recruit CCA communities, which would be more diverse than the single CCA treatment tested, and could be highly inductive^{30–32,62}. As in Turnlund et al.,⁹⁴ sequencing the cryptic taxa on more and less inductive fragments may provide further insights into key inductive and inhibitory communities.”
46. Lines 498-500: See Evenson et al. 2021 Ecology
Response: Citation added.
47. Lines 506-507: This is unsubstantiated. Why is it important to do a chemical characterisation of rubble substrata to identify potential inducers?
Response: This sentence follows the statement that legacy chemical inducers within the calcareous matrices may be influencing high settlement on rubble, and thus we contend should be characterized. However, we added the argument in response #454 above, and have thus broadened the statement to say “highlight the importance of further characterising rubble substrates...”
48. Lines 512-513: This was also showed by Harrington et al. 2004, Price 2010, Doropoulos et al. 2012, Ritson-Williams et al. 2016, where Porolithon was much more limited in inductiveness compared to other CCA.

Response: Thank you. This sentence was edited to add the suggested references: “Abdul Wahab et al. ³² recently demonstrated species-specific preferences amongst a broad taxonomic cross-section of coral/CCA pairings, and *Porolithon* was not the most universal CCA cue, corroborating previous findings ^{27,29,95,96}.”

49. Lines 559-573: Solid paragraph

Response: Thanks!

50. Line 586: “will improve the spatial planning process” See Doropoulos & Babcock 2018 Front Ecol Env

Response: Thank you, the citation as been added.

51. Table S1: Also see Nozawa & Harrison 2005 Coral Reefs for *F. chinensis*, *G. aspera*, Doropoulos et al. 2018 Coral Reefs for *A. digitifera*, *A. millepora*, *G. retiformis*

Response: Thanks for alerting us to these references. *G. retiformis* reference from Doropoulos et al. 2018 was added, but *A. millepora* and *A. digitifera* were not as the larvae in that study didn’t demonstrate settlement until CCA was added, but it is possible they were competent prior to the addition of CCA (8.0 d) and the other references cited suggest earlier competency in *A. millepora*. The other references were added.

REVIEWERS' COMMENTS:

Reviewer #1 (Remarks to the Author):

The authors addressed all my concerns.

Reviewer #2 (Remarks to the Author):

The authors have conducted a very thorough response and, where necessary, altered the text in the MS based on my and the other Reviewer's comments. While I still do not agree with the 30% threshold approach for quantifying the cohort settlement probabilities, the authors have provided more detailed rationalisation of the approach and open access to data for it to be interrogated by readers. Regardless, as I stated in my initial review, it's a really good piece of work and will likely be highly cited by the scientific community.

I have a few very minor comments to be considered:

1. Abstract - "25 abundant and widely distributed". Some of those species are not abundant (i.e., *O crispa*).
2. Data analyses - "as either competent (1) or not competent (2) to settle,". Shouldn't the "2" be a "0"?
3. Figure 4. Why do these plots show up to 100% settlement for many taxa (i.e., *A hyacinthus*, *D matthaii*, *D pallida*, *M elephantotus*, *O crispa*) that isn't apparent in Supp Table 3?

EDITORIAL REMARKS

Your manuscript entitled "Larval precompetency and settlement behaviour in 25 Indo-Pacific coral species" has now been seen again by our referees, whose comments appear below. In light of their advice I am delighted to say that we are happy, in principle, to publish a suitably revised version in Communications Biology under the open access CC BY license (Creative Commons Attribution v4.0 International License). We therefore invite you to revise your paper one last time to address the remaining concerns of our reviewers. At the same time we ask that you edit your manuscript to comply with our format requirements and to maximise the accessibility and therefore the impact of your work.

Response: Please see responses to editorial and reviewer comments below. We have edited the manuscript to comply with format requirements and maximise accessibility.

Please see the attached document for editorial requests for the final version (.docx file). Please ensure a completed version of this file is uploaded as a Related Manuscript with your final submission.

Response: The document is complete and uploaded as a related manuscript.

Please review our final submission file checklist to ensure all necessary files are present with your final submission and to avoid delays in accepting your manuscript. For your reference, a style and formatting guide is available here and includes all of our style requirements.

Response: The checklist has been reviewed and we believe we have followed all guidelines. I note that feature images have been submitted for consideration but an image license has not, due to broken links. I am happy to submit a license if an image is chosen.

An updated editorial policy checklist that verifies compliance with all required editorial policies must be completed and uploaded with the revised manuscript. All points on the policy checklist must be addressed; if needed, please revise your manuscript in response to these points. Please note that this form is a dynamic 'smart pdf' and must therefore be downloaded and completed in Adobe Reader. <https://www.nature.com/documents/nr-editorial-policy-checklist.pdf>

Response: The editorial policy checklist is complete and included in the revision.

REVIEWERS' COMMENTS:

Reviewer #1 (Remarks to the Author):

The authors addressed all my concerns.

Response: We thank the reviewer for their time in evaluating our revision.

Reviewer #2 (Remarks to the Author):

The authors have conducted a very thorough response and, where necessary, altered the text in the MS based on my and the other Reviewer's comments. While I still do not agree with the 30% threshold approach for quantifying the cohort settlement probabilities, the authors have provided more detailed rationalisation of the approach and open access to data for it to be interrogated by readers. Regardless, as I stated in my initial review, it's a really good piece of work and will likely be highly cited by the scientific community.

Response: We appreciate the reviewer's thorough evaluation and agree that open access data will allow the scientific community to make use of, and interrogate, the data. We believe it will be of great value to the community.

I have a few very minor comments to be considered:

1. Abstract - "25 abundant and widely distributed". Some of those species are not abundant (i.e., *O. cripsa*).

Response: We agree and have edited the text to say "...25 Indo-Pacific broadcast-spawning species...". Thanks for the pick-up.

2. Data analyses - "as either competent (1) or not competent (2) to settle,". Shouldn't the "2" be a "0"?

Response: Yes, thanks for the pick-up. The text was corrected.

3. Figure 4. Why do these plots show up to 100% settlement for many taxa (i.e., *A. hyacinthus*, *D. matthaii*, *D. pallida*, *M. elephantotus*, *O. cripsa*) that isn't apparent in Supp Table 3?

Response: In all the taxa listed by the reviewer, settlement reached 100% in at least one replicate of every species (see raw data in supplementary) and the mean + SE also reached at or very near 100%, as reported in Table S3 (*Ahya* reached 98 ± 2 on day 20; *Dmat* reached 89 ± 9 on day 7; *Dpal* reached 95 ± 5 on multiple days; *Mele* reached 89 ± 6 on day 12; *Ocr* reached 93 ± 4 on day 13). The models apply a cubic regression spline as a smoother; applying this smoother to the data, where at least one treatment reached at or near 100% mean settlement (with individual replicates reaching 100%), results in the models presented in figure 4. No change made.